# Evolution, structure and function of L-cysteine desulfidase, an enzyme involved in sulfur metabolism in the methanogenic archeon *Methanococcus maripaludis*

Sylvain Gervason [1,9], Paolo Zecchin [1,9], Elliot B. Shelton[2,9], Nisha He[1], Ludovic Pecqueur [1], Pierre Simon Garcia [3,7], Taiwo Akinyemi[2], Nadia Touati[4], Ornella Bimai[1], Christophe Velours[5,8], Jean-Luc Ravanat [6], Bruno Faivre[1], William B. Whitman [2], Marc Fontecave [1] & Béatrice Golinelli-Pimpaneau [1] ✉

The biosynthesis of sulfur-containing molecules, which play essential roles in cell metabolism, often relies on enzymes that mobilize sulfur from cysteine. The function of such enzyme, L-cysteine desulfidase CyuA, which catalyzes L-cysteine decomposition to pyruvate, ammonia, and hydrogen sulfide, remains incompletely understood. Here, we used phylogenetic, genetic, biochemical, spectroscopic, and structural approaches to connect molecular structure to cellular physiology and evolutionary history and elucidate CyuA's role in sulfur metabolism. We found that Methanococcales and several other archaeal lineages acquired CyuA via horizontal gene transfer from bacteria. In *Methanococcus maripaludis*, CyuA (MmCyuA) stimulates growth in sulfide-rich conditions and enables slow growth with cysteine as the sole sulfur source. Crystallographic and biochemical data reveal that MmCyuA binds a [4Fe-4S] cluster coordinated by three conserved cysteines; the fourth ligand is a nonconserved cysteine in the wild-type enzyme but is replaced by glycerol or ethylene glycol in a variant. These results enabled modeling of the enzyme–substrate complex, allowing us to propose a detailed mechanism for L-cysteine desulfuration by CyuA, potentially involving a transient [4Fe-5S] species to transfer sulfur from cysteine to various [4Fe-4S]-dependent tRNA sulfuration enzymes. These findings advance understanding of sulfur activation and trafficking related to biosynthetic pathways leading to sulfur-containing compounds.

Sulfur-containing molecules are widely distributed in nature and participate in essential biochemical reactions[1]. All organisms must therefore acquire sulfur from their environment for proper cellular function. In plants and many microorganisms, inorganic sulfate, which is the most abundant source of utilizable sulfur in the aerobic biosphere, is taken up and reduced to sulfide, which is incorporated into L-cysteine[2,3]. L-cysteine, in turn, is used for redox homeostasis, protein biosynthesis and serves as the primary source of sulfur for the biogenesis of L-methionine and sulfur-containing cofactors such as biotin, lipoate, thiamin, molybdopterin, coenzyme A[4–6], as well as [Fe-S]

clusters[7–9] and thionucleosides present within transfer RNAs (tRNAs)[5,10]. Sulfur mobilization from L-cysteine for biosynthesis of these molecules is catalyzed by various classes of essential pyridoxal-phosphate (PLP)-dependent L-cysteine desulfurases (CSD)[6,11,12]. The first step, which generates a persulfide attached to a catalytic cysteine of the enzyme, is followed by a complex network of sulfur-transfer reactions in which the sulfane sulfur atom of the persulfide is transferred to other sulfur-acceptor proteins.

In contrast to the cysteine metabolism described in bacteria and plants[2,3] different processes have been identified in archaea[13]. First, while in

most organisms, free cysteine is synthesized from O-acetylserine, in some methanogenic and other archaea, cysteine synthesis is carried out at the level of the aminoacylated tRNA before insertion into the polypeptide chain during protein translation[14–16]. Second, several archaea lack homologs of the nifS/iscS/sufS genes encoding CSDs[17,18], raising the still unsolved question of whether cysteine is the precursor of sulfur-containing biomolecules in these organisms. As a matter of fact, it was proposed that the sulfur atoms of [Fe-S] clusters do not come from cysteine in Methanococcus maripaludis[17], in contrast to organisms containing CSDs[19]. Instead, biosynthesis of these clusters was proposed to depend on inorganic sulfide, which is abundant in the medium of such archaea[17].

However, an enzyme that catalyzes desulfurization of cysteine into sulfide and 2-aminoacrylate, which is then converted into pyruvate and ammonia, has previously been characterized in Methanococcus jannaschii[20]. Such cysteine desulfidase activity could account for the observed release of free sulfide from cysteine in cell extracts, reported previously in M. maripaludis[17]. The low activity is consistent with the low levels of transcription and expression of cysteine desulfidase in this organism under normal growth conditions[21,22] (Supplementary Table 1). This L-cysteine desulfidase is a [4Fe-4S]-dependent enzyme that could also use L-selenocysteine as a substrate[20], raising the possibility that it participates in both sulfur and selenium metabolisms, at least under certain growth conditions. L-cysteine desulfidase has also been identified in several other organisms, including Escherichia coli[23–26], Salmonella enterica serovar Typhimurium[26], Moorella thermoacetica[27], the fish pathogen Yersinia ruckeri[28], and the gastrointestinal pathogen Clostridium difficile[29], and has been designated by various names (CyuA, YrbO, CdsB, YhaM, Csd) but only the enzyme from M. jannashii has been partly characterized biochemically[20]. Here, we chose to call this enzyme CyuA (for cysteine utilization A[26]). The physiological function of CyuA in such a wide variety of organisms and metabolic types remains poorly understood, in spite of being studied in depth in some bacteria, in particular when the encoding gene was expressed in the presence of exogenous cysteine[24,26,28,29]. However, a unified conclusion has not resulted from these studies since the enzyme could serve very different functions, such as cysteine detoxification[23,24], as a factor of virulence for C. difficile[29] and Y. ruckeri[28], or as cysteine utilization as a source of carbon/nitrogenenergy[25,27].

To better understand the role of CyuA, we investigated its taxonomic distribution and evolution, as well as its physiological relationship with cysteine metabolism and biochemical properties, and solved its first crystal structures using as a model the enzyme from M. maripaludis, a strictly anaerobic methanogenic archaeon that is common in salt water marshes[30,31], belonging to the class of archaea lacking a CSD[17,32]. Our study provides the following informations: (i) CyuA is mainly present in Terrabacteria but was transferred to some archaea such as Methanococcales via horizontal gene transfer (ii) CyuA is important for cellular growth of M. maripaludis with sulfide as the sole sulfur source and it enables this archaeon also to slowly grow with cysteine as the sole sulfur source; (iii) the crystal structure analysis of L-cysteine desulfidase allows us to propose a detailed catalytic mechanism for this enzyme family; (iv) this [4Fe-4S]-dependent enzyme uses its cluster to mobilize sulfur from L-cysteine and transfer it to support tRNA thiolation, suggesting that, in all organisms where it is found, this enzyme plays a crucial role in sulfur metabolism and has an important in vivo function.

## Results
### CyuA emerged from bacteria and has been restrained to specific organisms

Previous phylogenetic analysis suggested that CyuA was not present in the Last Universal Common Ancestor (LUCA)[33] and was specific to obligate and facultative anaerobes, suggesting an anaerobic function[20,26,28,29]. To deepen the taxonomic distribution of CyuA and specify its evolutionary history, we first identified the CyuA homologues in a large database encompassing more than 10,000 prokaryotic proteomes assembled in a previous analysis[9]. CyuA is homologous to [4Fe-4S]-dependent L-serine dehydratases (LSD)[34], named SdaA/B in the UniProt database, but it is also a

distant homologue of methyl-accepting chemotaxis proteins (Supplementary Fig. 1 and Supplementary Data 1–7). The distribution of CyuA is uneven in prokaryotes (Supplementary Fig. 2A). In Bacteria, CyuA is mostly present in Terrabacteria, one of the two major bacterial clades—notably in Wallbacteria/Riflebacteria, Firmicutes (Bacillota), Synergistetes, Thermotogae, and less common in the other clade of Gracilicutes (Spirochaetes, Schekmanbacteria, Deferribacteres, Proteobacteria, Pseudomonadata, etc.). The Terrabacteria are predominant, and their sequences from the same phylum are clustered, in comparison to Gracilicutes. In Archaea, CyuA is only found in Methanococcales and a few Methanocellales, Methanobacteriales, Lokiarchaeota and Woesarchaeota. Interestingly, CyuA is mainly present in organisms possessing the SdaA/B LSDs (Supplementary Fig. 2B) and in those possessing CSDs (Supplementary Fig. 3 and Supplementary Data 8). Methanococcales are an exception among prokaryotes, as they possess CyuA but no CSD (Supplementary Fig. 3). A phylogeny, based on a subsample of homologues and rooted using SdaA/B as an outgroup, shows that archaeal sequences are scattered within bacterial sequences (Fig. 1) and suggests that CyuA emerged in Terrabacteria and was later horizontally transferred to some Gracilicutes and Archaea. More specifically, CyuAs of Methanococcales are branching with two phylogenetically distant bacteria, Schekmanbacteria (Gracilicutes, sampled in fresh water) and Catenibacterium mitsuokai (Firmicutes, Terrabacteria, isolated from human faeces), suggesting that Methanococcales acquired CyuA through a series of horizontal gene transfers originating from some ancestors of these bacteria.

Next, we investigated whether the presence of CyuA is associated with specific environmental niches. In particular, we explored whether its emergence might be linked to oxygen exposure. To infer oxygen exposure from genomic data, we screened for five previously established marker genes that reliably distinguish aerobes from anaerobes[35], along with two additional genes - encoding catalase and cytochrome bo3 ubiquinol oxidase subunit I —used as proxies for aerobic metabolism[35] (Supplementary Fig. 4A). Our analysis revealed that the majority of organisms harboring CyuA lack all seven of these markers (1025 out of 1193 in the full dataset, 44 out of 51 in the homogeneously sampled subset). This suggests that CyuA is predominantly found in organisms not exposed to oxygen.

Further investigation into the habitats of CyuA-encoding organisms showed a strong enrichment in host-associated environments, particularly the animal gut, compared to the overall dataset (Supplementary Fig. 4B and Supplementary Fig. 4C). The remaining habitats were distributed between free-living natural environments—such as the deep biosphere, hydrothermal vents, marine, and soil ecosystems - and engineered or anthropogenic settings designed to support microbial life.

Taken together, these findings support the idea that CyuA is associated with anaerobic metabolism and is especially prevalent in organisms adapted to host-associated and oxygen-limited environments.

### Deletion of the gene encoding CyuA reduces growth of M. maripaludis

To examine the biological significance of CyuA, we chose to investigate its role in the archaeon M. maripaludis as a model. For that purpose, the mmp1468 gene encoding CyuA was deleted in M. maripaludis. The growth of the wild-type M. maripaludis S2 and the Δmmp1468 deletion mutant strains was compared with optimal (2.0 mM) or limiting (0.4 mM) concentrations of $Na_2S$. Indeed, Methanococcales usually live in an environment containing a high level of sulfide (2.5-10 mM, depending on the season[36]) and are usually cultivated in the presence of 2 mM $Na_2S$, with formate or a mixture of $H_2$ plus $CO_2$ serving as the carbon and energy sources[30,31]. Accordingly, we observed that 2 mM concentrations of $Na_2S$ were required for good growth of M. maripaludis under our conditions (Fig. 2A). The S2 strain clearly exhibited growth advantages over the Δmmp1468 deletion mutant when grown in rich formate medium for both concentrations of $Na_2S$, in terms of growth rate, cell yield, and lag time (Fig. 2A, Supplementary Table 2). For instance, in the presence of 2.0 mM $Na_2S$, the S2 strain had an average growth rate of 0.32 h$^{-1}$ (2.2 h doubling time), reached a final

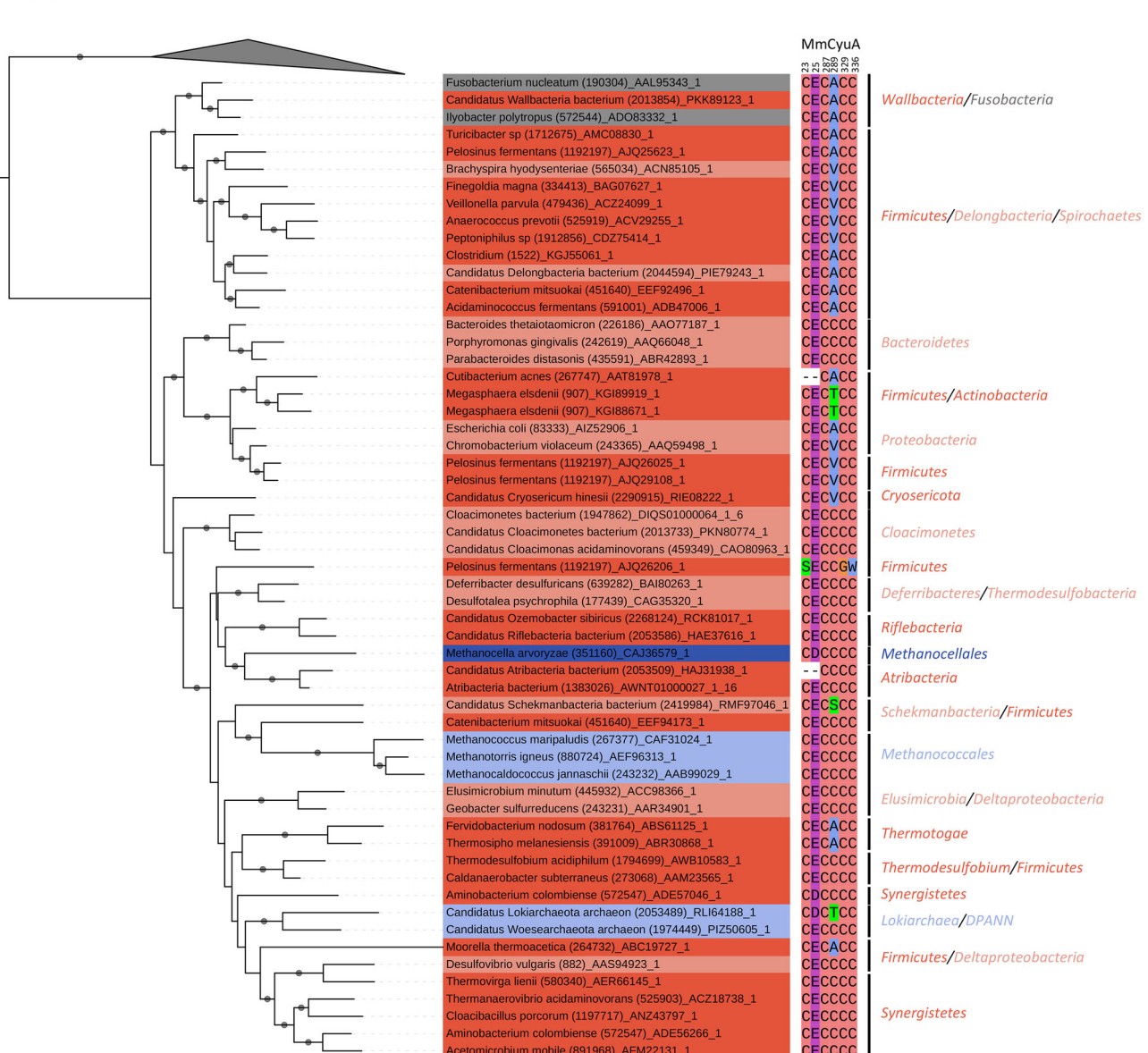

**Fig. 1 | Phylogeny of CyuA.** Phylogeny of CyuA (191 sequences, 236 amino-acid positions, IQ-TREE, LG + G4). The tree was rooted using SdaA/B sequences as an outgroup, collapsed as gray triangle. The dots at branches correspond to ultrafast bootstrap values ≥ 95%. The scale bar represents the average number of substitutions per site. The leaves are colored as in Supplementary Fig. 2 (dark red: Terrabacteria, light red: Gracilicutes, gray: Fusobacteria, light blue: Archaea cluster I, dark blue:

Archaea cluster II). The name of the organism is followed by the taxonomic ID (in parenthesis) and the protein accession number. The major phyla/clades are indicated on the right. The conservation of the six residues mentioned in the text is shown, with the numbers corresponding to the positions in the sequence of MmCyuA. The sequence alignment files are available as Supplementary Data 1–5 and the tree file is available as Supplementary Data 6.

optical density (OD) of around 1, and had a lag time of 19 h. Under the same conditions, $\Delta mmp1468$ had a growth rate of 0.23 h$^{-1}$ (3 h doubling time), reached a final OD of about 0.84, and had a lag time of 26 h (Fig. 2A, Supplementary Table 2). Both the wild-type and mutant strains grew poorly with 0.4 mM Na$_2$S, but the growth difference between the two strains was also very clear (Fig. 2A, Supplementary Table 2). Altogether, the relatively poor growth of the mutant, in the presence of both low and high levels of sulfide, suggests that *M. maripaludis* CyuA (MmCyuA) plays an important role in metabolism even when sulfide is not limiting growth.

**MmCyuA protects *M. maripaludis* against inhibition by cysteine**
We then investigated the potential function of MmCyuA as a protective mechanism for *M. maripaludis* to cope with high concentrations of L-cysteine. The addition of high concentrations of cysteine to the sulfide-containing medium inhibited growth of the S2 strain (Fig. 2B). Although

minimal changes in the growth parameters were observed at 1 mM cysteine, increasing the concentration of cysteine up to 10 mM decreased the growth rate and increased the lag time (Fig. 2B, Supplementary Table 2). With 2.0 mM Na$_2$S, 10 mM cysteine decreased the growth rate to 0.16 h$^{-1}$ and increased the lag time to 42 h (Supplementary Table 2). The growth inhibition effect was exacerbated by reducing the concentration of Na$_2$S from 2.0 to 0.4 mM, for which growth was almost completely abolished. The growth inhibition of the $\Delta mmp1468$ strain by cysteine followed a similar pattern to that of the wild-type S2 strain but was much more severe at both concentrations of Na$_2$S (Fig. 2B, Supplementary Table 2), indicating that MmCyuA protects *M. maripaludis* against growth inhibition by cysteine.

**MmCyuA is essential for cysteine utilization by *M. maripaludis***
In the few previous reports investigating the physiological role of CyuA in bacteria, the encoding gene was induced in the presence of exogenous

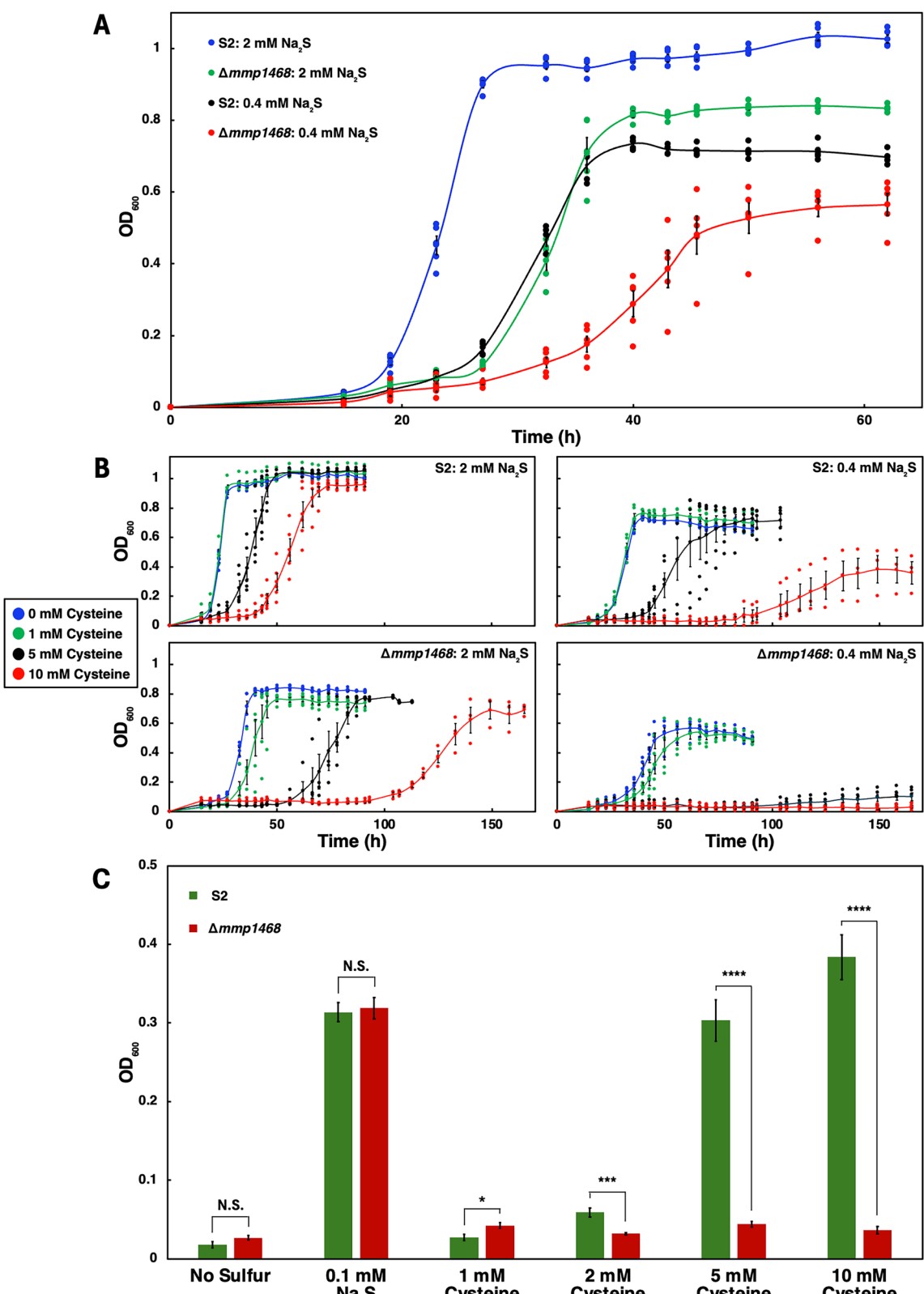

**Fig. 2 | Growth of the *M. maripaludis* S2 and Δ*mmp1468* strains in the presence of Na₂S and/or cysteine. A** The *mmp1468* gene is important for growth of *M. maripaludis* in the presence of sodium sulfide. The S2 and Δ*mmp1468* strains were grown to stationary phase in rich formate media in the presence of 2.0 mM or 0.4 mM Na₂S. **B** Cysteine inhibits growth of the S2 and Δ*mmp1468* strains grown in the presence of 2.0 mM or 0.4 mM Na₂S. **C** The *mmp1468* gene enables *M. maripaludis* to use cysteine only as the sole sulfur source for growth. The final OD of the S2 and Δ*mmp1468* strains grown under varied sulfide and cysteine conditions are shown. The inoculum per 5 mL culture was $10^5$ cells in (**A**, **B**) and $10^7$ cells in (**C**). The

individual replicate data points are shown as unlined marker points; the means are shown as smooth lined markers with the standard error of the mean (SEM) in (**A**, **B**). Where error bars are not seen, the SEM was smaller than the marker diameter. By a two-sample t-test assuming equal variances, the difference in final OD between the S2 and the Δ*mmp1468* strains was determined to be significant when grown on 2, 5, and 10 mM L-cysteine as the sole sulfur source. The *p*-values were $7.98 \times 10^{-4}$, $5.50 \times 10^{-6}$, and $1.00 \times 10^{-6}$, respectively. Additional testing details are available in the Supplementary data 9 and 10. *** = $p < 0.001$, **** = $p < 0.0001$. N.S.: non significant.

cysteine[24,26–29]. We then wondered whether or not L-cysteine could serve as the sole sulfur source for cellular growth of *M. maripaludis* and whether MmCyuA could be involved in sulfur mobilization from cysteine. Since both the wild-type and mutant strains failed to grow with 1–10 mM cysteine as the sole sulfur source using a low inoculum, we increased the load of the inoculum 100-fold to provide sufficient initial biomass to allow growth under stress conditions, using either sulfide at low concentration (0.1 mM) or cysteine (1–10 mM) as the sulfur source (Fig. 2C, Supplementary Table 2). In this medium, the abiotic formation of sulfide from 1 mM cysteine was below the limit of detection. When the S2 strain was grown in the absence of sulfide, with either 5 or 10 mM cysteine, the absorbance increased to 0.30 after an 18-day lag with a doubling time of 19 h, or to 0.38 after a 10-day lag with a doubling time of 26 h, respectively (Fig. 2C, Supplementary Table 2). In contrast, the lag time was less than ~18 h when sulfide was used as the sole sulfur source (Supplementary Table 2). Thus, although strain S2 could utilize cysteine as the sole sulfur source, the growth was very slow. Importantly, MmCyuA was essential for cysteine utilization since the Δ*mmp1468* mutant could not grow in the absence of sulfide, even with 10 mM cysteine in the growth medium (Fig. 2C).

## MmCyuA can bind a [4Fe-4S] cluster and is dimeric

MmCyuA was overexpressed in *Escherichia coli* with an N-terminal histidine tag and purified under aerobic conditions using nickel affinity chromatography. The tag was then removed using the human rhinovirus 3C (H3C) protease, and the protein was further purified by size-exclusion chromatography (Supplementary Fig. 5A, B). The so-called 'as-purified' MmCyuA protein possessed a brownish color associated with a UV-visible spectrum exhibiting bands at around 320, 410, and 460 nm, characteristic of an Fe-S cluster (Supplementary Fig. 5C, dashed line)[37]. Quantification of iron and sulfur contents gave $1.1 \pm 0.2$ mol Fe and $1.2 \pm 0.2$ mol S per monomer. These data support the presence of an iron-sulfur cluster-binding site in MmCyuA. The low Fe content reflects the oxygen sensitivity of these clusters.

After treatment with dithionite and EDTA to remove the residual cluster (Supplementary Fig. 5C, dotted line), the protein was named apo-MmCyuA. Size-exclusion chromatography with multi-angle light scattering (SEC-MALS) analysis of apo-MmCyuA indicated that it was mainly a dimer (Supplementary Fig. 5D). The cluster was then reconstituted anaerobically by treating the protein with a 5-molar excess of ferrous iron and sodium sulfide. The protein was then purified on a Superdex 200 gel filtration column (Supplementary Fig. 6A, B), also under anaerobic conditions, leading to a homogenous brownish protein containing a [Fe-S] cluster that was subsequently called holo-MmCyuA. SEC-MALS analysis of holo-MmCyuA, using the correction for samples that absorb the laser light of the SEC-MALS device[38], indicated that it forms a dimer (measured molar mass of $80.8 \pm 2.1$ kDa and theoretical mass of monomer of 43.7 kDa) (Supplementary Fig. 6C).

Quantification of the protein-bound iron and sulfur content gave $3.8 \pm 0.2$ Fe and $3.4 \pm 0.3$ S per holo-MmCyuA monomer, consistent with the presence of one [4Fe-4S] cluster per monomer. Accordingly, the UV-visible spectrum of purified holo-MmCyuA displayed a broad absorption band at around 400 nm that is characteristic of the presence of a [4Fe-4S]$^{2+}$ cluster (Supplementary Fig. 7A, solid line)[37]. Addition of dithionite to holo-MmCyuA led to a rapid decrease of the intensity of the 400 nm absorption band, suggesting a fast reduction of the cluster (Supplementary Fig. 7A, dashed line). Further analysis by Electron Paramagnetic Resonance (EPR) spectroscopy (Supplementary Fig. 7B)[37] showed that holo-MmCyuA was EPR silent, in agreement with a cluster in a $S = 0$ state and excluding the presence of $S = 1/2$ paramagnetic clusters such as [4Fe-4S]$^+$ or [3Fe-4S]$^+$. Upon reduction with dithionite, the EPR spectrum exhibited an axial signal characteristic of a $S = 1/2$ [4Fe-4S]$^+$ cluster with $g_{//}=2.03$ and $g_{\perp}=1.93$. The variation of the EPR spectrum with temperature confirmed the [4Fe-4S]$^+$ state of the cluster (Supplementary Fig. 7C).

## The [4Fe-4S] cluster is required for cysteine desulfidase activity

The cysteine desulfidase activity of MmCyuA was assayed using L-cysteine as the substrate by monitoring pyruvate formation after derivatizing with 2,4-dinitrophenylhydrazine using visible spectroscopy, as reported[39]. The cluster was required for catalysis since only holo-MmCyuA, and not apo-MmCyuA, exhibited catalytic activity (Fig. 3A). Na$_2$S (up to 5 mM) did not decrease the pyruvate formation rate, indicating that product inhibition was not significant (Fig. 3B). Dithiothreitol (DTT) was an inhibitor since 5 mM DTT decreased the activity by about 45% (Fig. 3B) and was thus omitted in subsequent assays. Holo-MmCyuA showed typical Michaelis-Menten saturation kinetics with L-cysteine as a substrate with $K_m$ and $k_{cat}$ values of $0.25 \pm 0.05$ mM and $2.6 \pm 0.1$ s$^{-1}$, respectively (Fig. 3A and Supplementary Fig. 8A). Interestingly, holo-CyuA could also use L-selenocysteine as a substrate and followed Michaelis-Menten kinetics with $K_m$ and $k_{cat}$ values of $0.21 \pm 0.2$ mM, $1.9 \pm 0.1$ s$^{-1}$, respectively (Supplementary Fig. 8B). Moreover, L-serine behaved as a competitive inhibitor of holo-MmCyuA since the Lineweaver burk plots at different serine concentrations intersected on the y-axis (Supplementary Figs. 8C). The Dixon plot gave a $K_i$ of $15 \pm 0.3$ μM (Supplementary Fig. 8D).

## MmCyuA can transfer sulfur from L-cysteine to tRNA-sulfurating enzymes in vitro

To know whether MmCyuA could mobilize sulfur from L-cysteine for tRNA thiolation[40], we tested the activity of [4Fe-4S]-dependent U34-tRNA and U8-tRNA sulfur transferases from *M. maripaludis*[41,42], named MmNcsA and MmTtuI, respectively. The sulfur donor was either inorganic sulfide or L-cysteine in the presence of MmCyuA (Supplementary Fig. 9). Our results confirmed that the [4Fe-4S] clusters of MmNcsA and MmTtuI are absolutely required for tRNA sulfuration using sulfide as a sulfur source, as reported[41,42]. Remarkably, in the absence of sulfide, MmCyuA, together with L-cysteine, supported tRNA sulfuration, with the [4Fe-4S] cluster of MmCyuA being absolutely essential.

## Holo-MmCyuA has a dimeric architecture

Holo-MmCyuA was crystallized under anaerobic conditions within a glove box. Two types of crystals were obtained in the space group $P2_12_12_1$, with two molecules in the asymmetric unit, which diffracted to 2.4–2.9 Å resolution (Crystals 1 and 2, Table 1). The structure was solved by single wavelength anomalous diffusion (SAD) using data collected near the Fe K-edge on Crystal 2. Indeed, an initial anomalous map at 5 Å resolution allowed the positioning of two [Fe-S] clusters in the asymmetric unit, whereas molecular replacement (MR) using the most closely related protein in the PDB, L-serine dehydratase from *Legionella pneumophila* called LpLSD (PDB code 4RQO, 12.4% sequence identity with MmCyuA; Supplementary Fig. 10A)[34], led to manual building of an incomplete model. Finally, this model and the anomalous data were combined in CRANK2[43] using the MR-SAD mode to successfully solve the structure.

The two molecules in the asymmetric unit formed a dimer (Fig. 4A), with the interface provided by hydrophobic residues (Supplementary Fig. 11A). Analysis of the contacts between the two monomers with the 'Protein interfaces, surfaces and assemblies' (PISA) software[44] indicated that this packing, with a buried surface area of 3680 Å$^2$, corresponded to the dimer in solution. Each monomer in the asymmetric unit was constituted by a N-terminal structural domain (residues 24–145), formed by a central antiparallel β-sheet surrounded by α-helices, and a mostly helical C-terminal structural domain (residues 157–378) (Supplementary Fig. 11B, C). The two domains were connected by a linker (residues 146–156) that could not be completely traced due to the lack of interpretable electron density. Interestingly, the formation of an antiparallel β-sheet between strand β1 (residues 17–21) and strand β9 (residues 230–234) resulted in the N-terminal extremity of MmCyuA (residues 1 to 43) becoming part of the C-terminal structural domain (Supplementary Fig. 11B). The structures of the two monomers were nearly identical to each other (Supplementary Fig. 11C).

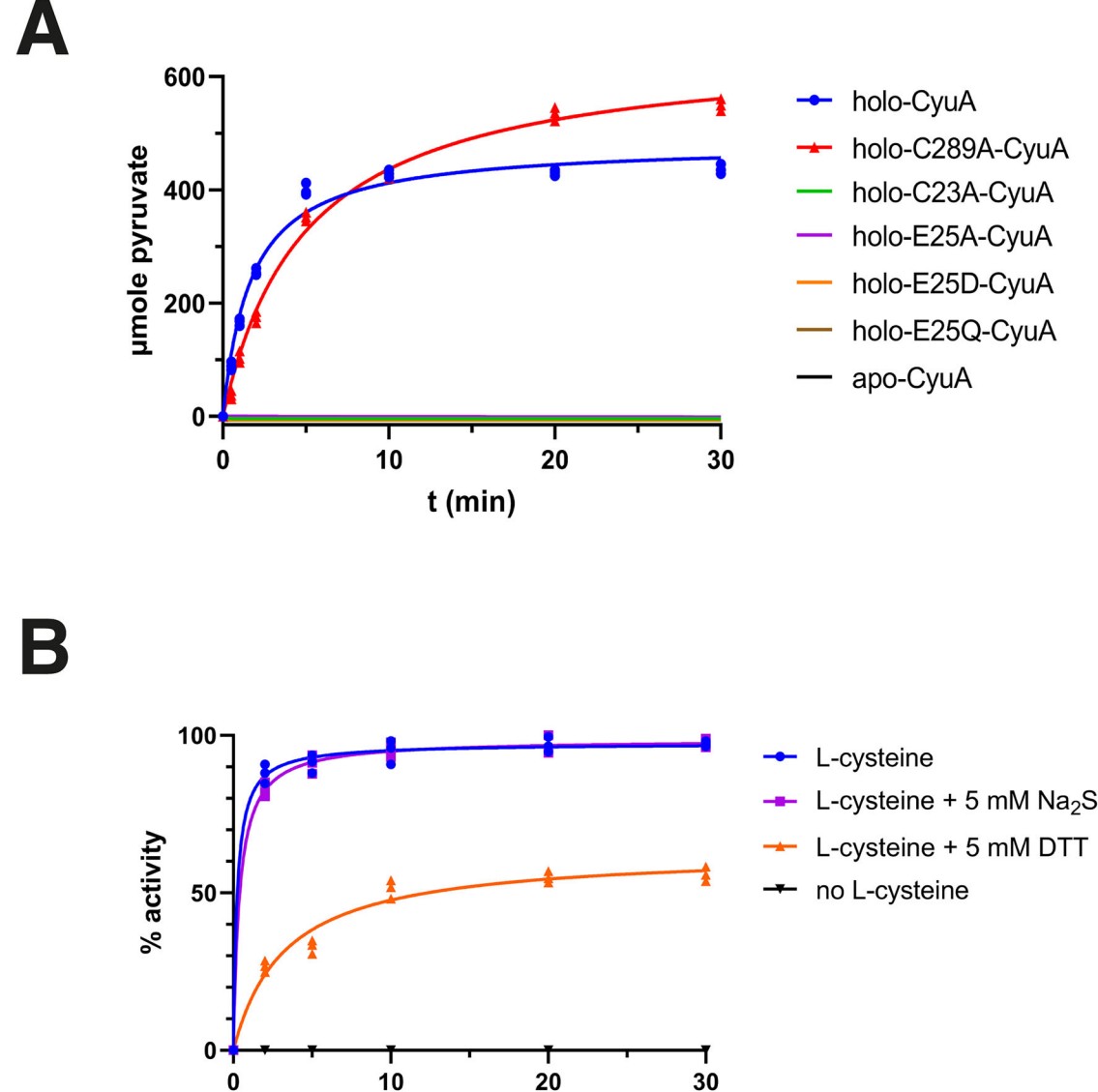

**Fig. 3 | Cysteine desulfidase activity of MmCyuA.** The various MmCyuA proteins (1 μM) were incubated in the presence of 1 mM cysteine in 25 mM HEPES pH 7.5, 0.15 M NaCl. **A** In contrast to holo-MmCyuA and holo-C289A-MmCyuA, apo-MmCyuA and the C23A and Glu25 variants of holo-MmCyuA were inactive. **B** The activity of holo-MmCyuA was inhibited by 5 mM DTT but not by 5 mM Na2S. The data shown are mean values based on 3 different experiments.

### The [4Fe-4S] cluster is bound by four cysteines in wild-type MmCyuA

The 2Fo-Fc map at 2.94 Å resolution (Supplementary Fig. 12A) and the anomalous maps from data recorded near the iron K-edge energy (Supplementary Fig. 12B) of holo-MmCyuA crystallized in the absence of ligand (Crystal 1) indicated that one [4Fe-4S] cluster was bound to each molecule in the asymmetric unit. In both molecules, the electron density was best fitted by one [4Fe-4S] cluster with a 65% occupancy, coordinated by four cysteines (Cys287, Cys289, Cys329 and Cys336) (Supplementary Fig. 12C). Among these, Cys287, Cys329 and Cys336 are strictly conserved (Supplementary Fig. 10B), with Cys287 and Cys329 belonging to flexible loops. In particular, residues A283-A286 and B283-B286 were not modeled due to the absence of electron density. Intriguingly, the nonconserved residue Cys289 (Supplementary Fig. 10B) was also a ligand of the cluster. In both molecules, besides the cysteines that coordinate the cluster, the closest neighbors to the cluster were Lys339, Asn239, Cys23, Glu25 and a sulfate ion (Fig. 4B). The sulfate ion, which occupied an anion binding site, formed ionic interactions with Arg223 and hydrogen bonds with the SH group of a remote cysteine (Cys23) and to the side chain and main chain nitrogen atoms of Asn239 (Fig. 4B).

### The crystal structure of the [2Fe-2S]-MmCyuA in complex with MES confirms the location of an anion binding site

Attempts were made to co-crystallize holo-MmCyuA with the competitive inhibitor serine (10 mM). The crystals, obtained also in space group P2₁2₁2₁, diffracted at 2.4 Å resolution and possessed two molecules in the asymmetric unit (Table 1, crystal 2), which differed by the orientation of the N-terminal domains relative to the catalytic domains (Supplementary Fig. 13A). The anomalous maps from data recorded near the iron K-edge energy were best fitted with a [2Fe-2S] cluster bound to the four cysteines (Cys287, Cys289, Cys329 and Cys336), with an occupancy of 90% (Supplementary Fig. 13B and Supplementary Fig. 13C). In both molecules, the electron densities observed near the cluster did not fit with a serine molecule and were instead attributed to 2-(N-morpholino)ethanesulfonic acid (MES) in molecule A (Supplementary Fig. 13D) and to a chloride ion in molecule B (Supplementary Fig. 13E), both present in the crystallization solution at a

**Table 1 | Data collection and refinement statistics[a]**

| Data collection [a] | Crystal 1 [4Fe-4S]-CyuA | Crystal 2 [2Fe-2S]-CyuA + MES | Crystal 3 [4Fe-4S]-C289A-CyuA + ethylene glycol | Crystal 4 [4Fe-4S]-C289A-CyuA + glycerol |
|---|---|---|---|---|
| PDB code | 9FYI | 9FSL | 9HE2 | 9HE0 |
| Wavelength (Å) | 0.980 | 1.73891 | 0.979 | 0.979 |
| Beam line | PX2 | PX2 | PX1 | PX1 |
| Space group | $P2_12_12_1$ | $P2_12_12_1$ | P 3$_2$ 2 1 | $P$ 3$_2$ 2 1 |
| Cell dimensions | | | | |
| $a, b, c$ (Å) | 67.0, 79.0, 156.2 | 64.2, 77.2, 154.0 | 74.6, 74.6, 251.1 | 73.7, 73.7, 251.4 |
| $\alpha, \beta, \gamma$ (°) | 90, 90, 90 | 90, 90, 90 | 90, 90, 120 | 90, 90, 120 |
| Resolution (Å) [b] | 78.12–2.94 (3.19–2.94) | 76.98–2.42 (2.62–2.42) | 19.91–3.74 (4.04–3.74) | 19.9–3.18 (3.42–3.18) |
| $R_{-meas}$ (%)[b] | 11.9 (197.0) | 11.4 (415.6) | 15.2 (168.5) | 46.1 (449.6) |
| $I/\sigma(I)$ [b] | 11.5 (1.2) | 16.5 (1.5) | 9.0 (1.3) | 6.5 (1.3) |
| $CC_{1/2}$ [b] | 99.6 (52.0) | 99.8 (75.0) | 99.8 (67.9) | 99.6 (50.3) |
| Completeness (%)[b] | | | | |
| Spherical Ellipsoid | 82.5 (19.7) 92.8 (43.0) | 77.2 (18.2) 92.4 (55.7) | 81.8 (18.6) 91.7 (37.1) | 81.3 (21.4) 91.6 (51.0) |
| Redundancy [b] | 10.9 (10.3) | 23.3 (24.2) | 11.0 (10.6) | 19.6 (20.2) |
| $B$ Wilson (Å$^2$) | 113.9 | 80.4 | 171.0 | 89.5 |
| Refinement | | | | |
| Resolution (Å) | 70.62–2.94 | 76.98–2.42 | 19.91–3.74 | 19.9–3.18 |
| No. reflections | 15210 | 23186 | 7385 | 11320 |
| $R_{work}$ / $R_{free}$ | 0.222/0.261 | 0.232/0.276 | 0.217/0.263 | 0.209/0.239 |
| No. atoms | 5640 | 5619 | 5979 | 6033 |
| Protein | 5614 | 5523 | 5954 | 6005 |
| Fe, S | 8,8 | 4, 4 | 8,8 | 8,8 |
| Ligands | / | 12 (MES) | 8 (Ethylene glycol) | 12 (Glycerol) |
| Water | / | 19 | 1 | / |
| Ions | 10 (sulfate) | 1 (Cl$^-$) | / | / |
| B factors (Å$^2$) | | | | |
| Protein | 119.5 | 89.7 | 162.96 | 75.68 |
| Fe, S | 152.0 | 93.4 | 146.52 | 67.95 |
| Ligands | / | 65.6 | 135.53 | 61.77 |
| Water | / | 65.1 | 103.26 | / |
| Ions | 116.5 | 101.79 | / | / |
| R.m.s. deviations | | | | |
| Bond lengths(Å) | 0.007 | 0.007 | 0.003 | 0.002 |
| Bond angles (°) | 0.89 | 0.92 | 0.55 | 0.49 |

[a]one crystal for each structure.
[b]Values in parentheses are for highest-resolution shell.

concentration of 100 mM. Thus, the structure corresponding to Crystal 2 was named [2Fe-2S]-CyuA + MES. The presence of two different ligands near the active sites of molecules A and B leads to slightly different positioning of the clusters (Supplementary Fig. 13E). The sulfonate group of MES and the chloride ion were located at the same anion binding site occupied by the sulfate ion in the holo-MmCyuA structure.

## In the crystal structure of holo-C289A-MmCyuA, the [4Fe-4S] cluster is coordinated by the three conserved cysteines and a glycerol or ethylene glycol ligand

To test whether the nonconserved Cys289 was necessary for enzyme activity, it was substituted to alanine. After purification of the C289A-MmCyuA variant and cluster reconstitution (Supplementary Fig. 14 & Supplementary Table 3), holo-C289A-MmCyuA displayed a desulfidase activity similar to that of the wild-type enzyme (Fig. 3A), indicating that Cys289 is not required for cluster binding and catalysis. X-ray data were collected for crystals of the C289A-MmCyuA variant after cryoprotection with 25% ethylene glycol or after soaking with serine (100 mM) and cryo-protection with 25% glycerol. The crystals belonged to space group P3$_2$21, with two molecules in the asymmetric unit and diffracted at 3.74 and 3.18 Å resolution, respectively (Crystals 3 and 4, Table 1). The space group was different from that of the wild-type enzyme crystals, explaining differences between these structures (Supplementary Fig. 15A), especially in several loops (279–287, 323–333, 353–365 and the linker between the two domains; Supplementary Fig. 11B). In addition, comparison of the active sites revealed differences in the positioning of several amino acids next to the cluster such as Cys23, Glu25 and Arg223 (Supplementary Fig. 15B). The two molecules in the asymmetric unit of holo-C289A-MmCyuA were quite different (overall superposition of 1.272 Å over 333 Cαs) because of a different orientation of one domain relative to the other, as well as different conformations of several loops such as the loop connecting residues 136 to 157 (Supplementary Fig. 15C). In each molecule, the anomalous density maps near the Fe K-edge indicated the presence of one [4Fe-4S] cluster with full occupancy, bound by the three conserved cysteines only (Fig. 4C). Remarkably, an ethylene glycol molecule, coming from the cryoprotectant solution, was bound to the [4Fe-4S] cluster of holo-C289A-MmCyuA crystal 3, thus named holo-C289A-MmCyuA + ethylene glycol, as shown by the electron density map (Supplementary Fig. 15D). Despite the low resolution of 3.74 Å of this structure, it is likely that one hydroxyl group of ethylene glycol is coordinated to one Fe atom (named unique Fe atom) of the cluster, whereas the other one makes hydrogen bonds with the guanidinium group of Arg223. The three other Fe atoms were coordinated by Cys287, Cys329 and Cys336. Unexpectedly, for crystals soaked with 100 mM serine (crystal 4), diffracting at 3.18 Å resolution, the extra electron density around the unique Fe atom of the [4Fe-4S] cluster was better fitted with a glycerol molecule coming from the cryoprotectant solution rather than with a L-serine ligand. One terminal hydroxyl of the glycerol molecule was coordinated to that Fe atom, the second one was hydrogen bonded to the guanidinium group of Arg223, whereas the C2 hydroxyl group formed a hydrogen bond with the main chain nitrogen atom of Asn239 (Fig. 4D). The crystal 4 structure, thus named holo-C289A-MmCyuA + glycerol, together with that of crystal 3 with bound ethylene glycol, were used to model the binding of the cysteine substrate and the serine inhibitor to MmCyuA (Supplementary Fig. 15E). In these models, the SH group of cysteine and the OH group of serine are coordinated to the unique Fe atom of the [4Fe-4S] cluster, while the carboxylate group of the serine/cysteine ligands occupy the same position as the sulfonate group of MES in the MmCyuA/MES structure or the sulfate ion in the MmCyuA structure. The model obtained with cysteine likely mimics the enzyme-substrate complex initiating the catalytic cycle.

## Cys23 and Glu25 of MmCyuA are critical for the cysteine desulfidase activity

The structures of holo-MmCyuA and holo-C289A-MmCyuA showed that, among the residues located in the active site, Cys23 and Glu25 are the best suited to play a catalytic role. The side chains of Cys23 and Glu25 are the closest to the Cα atom of the modeled serine/cysteine ligands (4.5 and 5.4 Å, respectively) (Supplementary Fig. 15E). Glu25, which is highly conserved and replaced with an aspartate in a few genomes (Fig. 1 and Supplementary Fig. 10B), is the best candidate to play a role as a base in catalysis to abstract the Cα proton of the cysteine substrate. Cys23, which is located near the anion site and forms a hydrogen bond with one oxygen atom of the

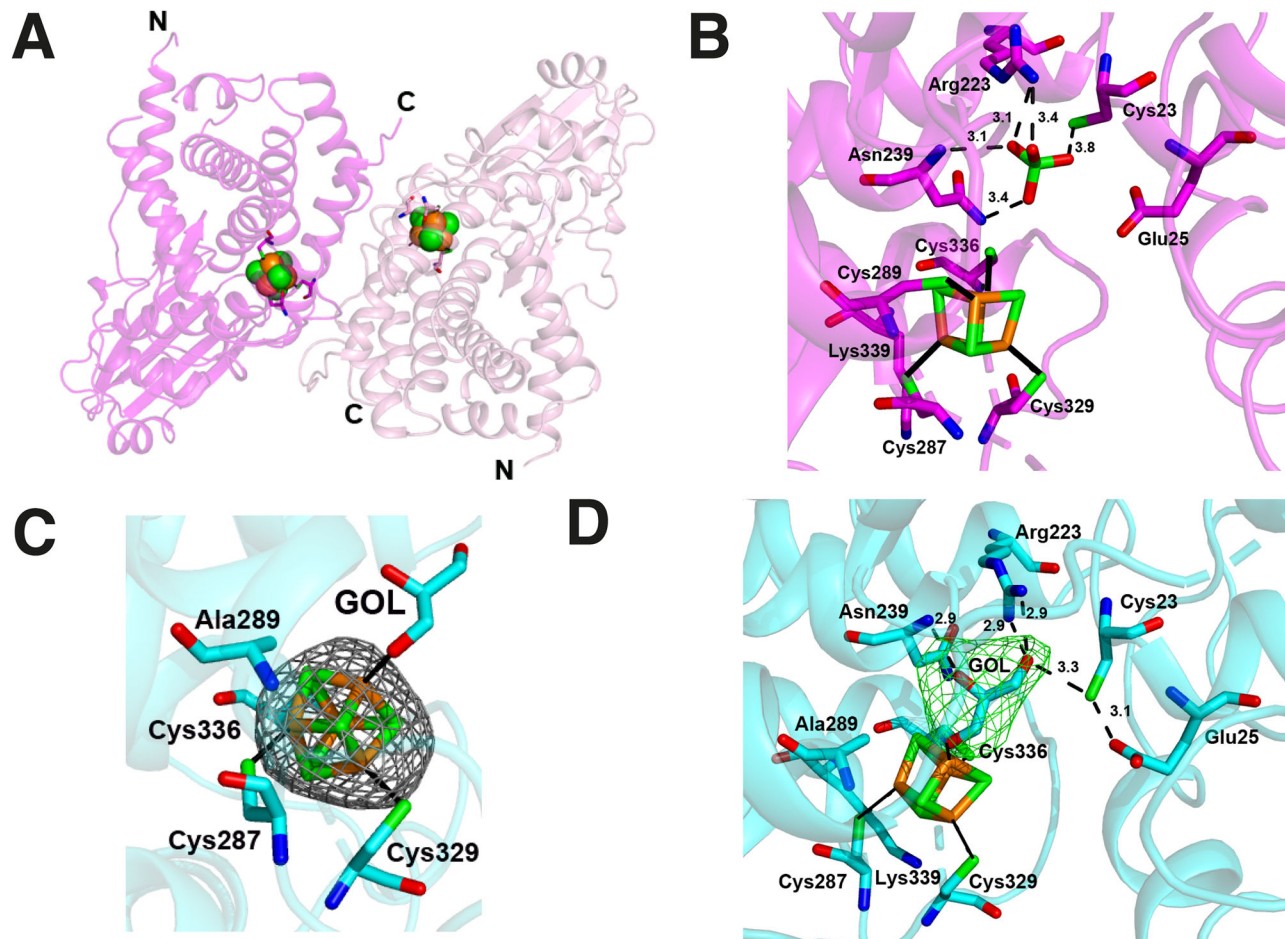

**Fig. 4 | Crystal structure of holo-MmCyuA. A** Overall structure of the holo-MmCyuA dimer (monomer A in magenta, monomer B in pink). The Fe and S atoms of the clusters are shown as orange and green spheres, respectively, and the cysteines that coordinate the cluster as sticks. N and C indicate the N and C termini, respectively. **B** Environment of the [4Fe-4S] cluster in holo-MmCyuA alone (molecule A) with a bound sulfate molecule that locates an anionic site. Dashed lines and numbers indicate the distances between atoms interacting within 4 Å. **C** Anomalous map contoured at 3.5 σ (in grey) for data collected near the K-edge of iron for the holo-C289A-MmCyuA crystal with bound glycerol (GOL). **D** Environment of the [4Fe-4S] cluster site in holo-C289A-MmCyuA with bound glycerol. A Fobs-Fcal map omitting the glycerol ligand (in green) is contoured at 3.5 σ.

carboxylate group of Glu25 in the holo-C289A-MmCyuA structure (Figure Supplementary Fig. 15D), is not strictly conserved since it is replaced by a valine, a serine or is absent in some CyuA sequences (Fig. 1 and Supplementary Fig. 10B). Nevertheless, present in most enzymes, Cys23 could play a role in catalysis by assisting the deprotonation of the carboxylate group of Glu25. To examine the function of residues Cys23 and Glu25, these residues were substituted to various amino acids. The C23A, E25A, E25Q and E25D variants were overexpressed and purified, then reconstituted with a [4Fe-4S] cluster, which led to a Fe content of 3.2–3.8 (Supplementary Fig. 14 and Supplementary Table 3). All the holo variants were completely inactive (Fig. 3A), indicating an important role for both Cys23 and Glu25 in the catalytic mechanism. Interestingly, these two residues are not conserved in the SdaA/B family (Supplementary Fig. 1), highlighting the difference in substrate specificity between L-cysteine desulfidases and L-serine dehydratases.

### The C-terminal domain of MmCyuA displays strong structural similarity with the catalytic domain of serine dehydratases

Analysis of the MmCyuA crystal structure with DALI (Distance matrix alignment)[45] indicated that the protein with known crystal structure that has the highest structural similarity was LpLSD[34] (PDB code 4RQO, Fig. 5) (Z-score of 23.5, rmsd = 3.8 Å for 324 aligned Cαs), despite the low sequence identity between the two proteins (Supplementary Fig. 10A). The dimeric interface of MmCyuA and LpLSD was also conserved (Fig. 5A).

Interestingly, the C-terminal catalytic domain of MmCyuA was not only structurally similar to that of LpLSD (Z-score of 12.6, rmsd = 2.5 Å for 155 aligned Cαs) but also to the catalytic domains of cis-aconitate decarboxylase (Z-score of 13.5, rmsd = 2.6 Å for 155 aligned Cαs, PDB code 7BR9)[46] and iminodisuccinate epimerase (Z-score of 13.0, rmsd = 2.3 Å for 153 aligned Cαs, PDB code 2HP3)[47], two enzymes that function without any cofactor, for which a two-base mechanism has been proposed. The LpLSD crystal structure showed that three iron atoms of the [4Fe-4S] cluster were coordinated with the three conserved cysteines and that the fourth iron atom was bound by the C-terminal cysteine (Cys458) (Fig. 5B)[48]. The superposition of the active sites of holo-C289A-MmCyuA and LpLSD showed that the glycerol ligand in the first structure occupies a position similar to that of Cys458 in the LpLSD structure (Fig. 5B). His61 of LpLSD, which likely acted as the catalytic base that abstracts the Cα proton of the serine substrate[34], is located similarly as Cys23/Glu25 in MmCyuA (Fig. 5B).

### Discussion

We have here provided the first exhaustive characterization of CyuA, exploring altogether its evolution, physiological role, enzymatic activity and structure. CyuA is an interesting enzyme because it lies at the crossroad of different major metabolisms due to its variety of substrates (cysteine or selenocysteine) and products. Indeed, because the final products, pyruvate, ammonia and sulfide/selenide, constitute potential direct major sources of carbon, nitrogen and sulfur/selenium, respectively, CyuA can fulfill several

**Fig. 5 | Comparison of the crystal structures of MmCyuA and LpLSD. A** Superposition of the dimers of holo-MmCyuA (monomers in pink and magenta, clusters as spheres) and LpLSD (monomers in green and forest green, clusters as sticks). The C-terminal catalytic domains of molecules A of MmCyuA alone and LpLSD were superposed using the SUPER option in PYMOL, with an rmsd of 4.3 Å for 128 aligned Cαs. **B** Molecules A of holo-C289A-MmCyuA in complex with glycerol (in cyan) and LpLSD (in forest green) were superimposed with an rmsd of 3.73 Å for 129 Cαs with the SUPER option in PYMOL.

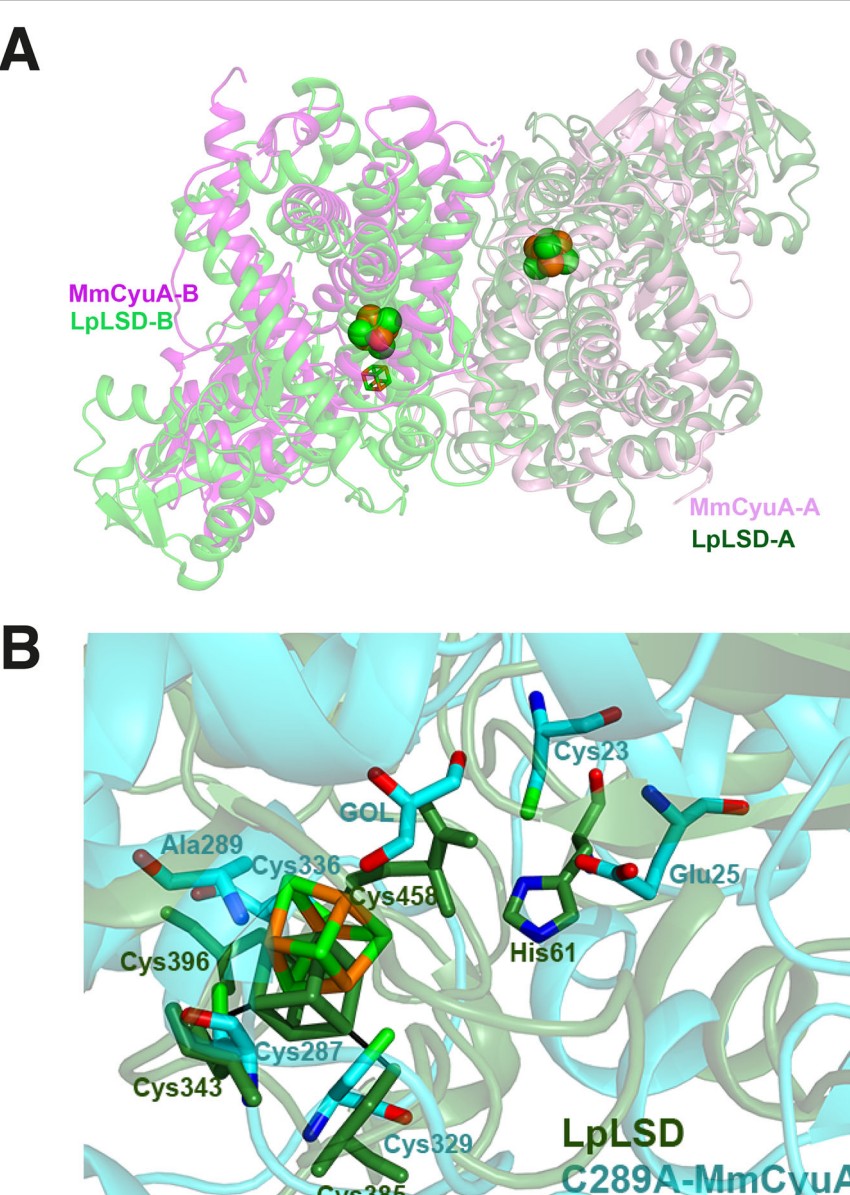

roles in the cell metabolism. First, CyuA can metabolize L-cysteine as a carbon/energy source. For example, in *M. thermoacetica*, the pyruvate produced by CyuA could be used directly for acetic acid fermentation[27], whereas the *cyuA* gene allowed some organisms, such as *E. coli* and *S. enterica*, to grow in the presence of cysteine as the sole carbon source[24,26]. Second, it was shown that the concerted action of CyuA and the cysteine importer CyuP allowed anaerobic *E. coli* to employ cysteine also as the sole nitrogen source[25]. Third, we now show here that CyuA can also participate in sulfur metabolism in organisms such as *Methanococcales* potentially in several ways.

Our phylogenetic results clearly confirm that most organisms possessing CyuA are anaerobes, supporting a role in anaerobic metabolism, and that it is not an ancestral protein, in agreement with a previous global analysis of the hypothetical genome of the LUCA[33]. Surprisingly, closely related lineages within the CyuA phylogeny inhabit markedly different environments, exemplified by *Methanococcales* and their sister lineages. The acquisition of CyuA in *Methanococcales* likely occurred in an anaerobic environment, where methanogens and donor bacteria coexisted, such as sediments or hydrothermal systems. This scenario illustrates a case of functional convergence, where unrelated lineages utilize and transfer the

same gene. It suggests that CyuA functions as a versatile metabolic module with various roles across diverse ecological contexts. This raises the question of what has driven the emergence and patchy distribution of CyuA during the diversification of the *Terrabacteria*. Its presence in specific ecological niches may be linked to a role in conferring tolerance to toxic levels of sulfide or cysteine. To explore this possibility, we investigated the function of MmCyuA in sulfur metabolism.

We report here the first homogeneous preparation of a holo-CyuA enzyme, as well as its spectroscopic and enzymatic characterization. Previously, the aerobically purified CyuA from *Methanocaldococcus janashii* (MjCyuA) was described as being probably trimeric and shown to contain a $[3Fe-4S]^+$ cluster with 2.7 mole of Fe per subunit[20]. Its catalytic activity was measured by converting the inactive $[3Fe-4S]^+$ cluster into the active $[4Fe-4S]^{2+}$ cluster using DTT in the incubation mixture. Unlike MjCyuA, after purification under aerobic conditions, as-purified MmCyuA behaved as a dimer, with each monomer containing a [2Fe-2S] cluster, a state of the cluster previously observed for several other [4Fe-4S]-dependent proteins when purified aerobically[49], presumably due to degradation of the cluster by oxygen. We showed that, after anaerobic cluster reconstitution, MmCyuA can bind one [4Fe-4S] cluster per subunit, which was essential for cysteine

**Fig. 6 | Proposed mechanism for [4Fe-4S]-dependent L-cysteine desulfidases CyuA.** In the resting state, the [4Fe-4S] cluster of MmCyuA is coordinated by four cysteinyl residues of the enzyme. Upon binding the L-cysteine substrate, the nonconserved cysteine would move away from the [4Fe-4S] cluster to enable binding of the SH group of the L-cysteine substrate to the unique Fe atom of the cluster. Deprotonation of the Cα atom of L-cysteine would then induce Fe-assisted C-S bond cleavage, generating 2-amino-acrylate and a potential [4Fe-5S] intermediate that would serve as the sulfur source in various reactions. Note that the protonation state of the L-cysteine substrate is unknown so that it could coordinate the [4Fe-4S] cluster in its thiolate form.

desulfidase activity. In addition to cysteine, selenocysteine was a substrate of MmCyuA, like for MjCyuA[20]. Selenocysteine is the most widely distributed biologically active form of selenium, which is an important micronutrient for several organisms[50]. Moreover, selenocysteine is a key amino acid residue in some essential proteins, and selenocysteine-containing proteins are widespread in some archaea, including *Methanococcales*[51], most of them being involved in methanogenesis[52–54]. Therefore, the activity of CyuA towards selenocysteine may be physiologically important in these organisms.

We also report the first crystal structures of a CyuA enzyme, in the holo-form, which showed that the three strictly conserved cysteines (Cys287, Cys329 and Cys336 in MmCyuA) were ligands of the [4Fe-4S] cluster, in agreement with the mutagenesis analysis of the equivalent cysteines in MjCyuA[20]. In addition, in wild-type holo-MmCyuA, the non-conserved cysteine Cys289, although not essential for activity, was also bound to the [4Fe-4S] cluster. In fact, the slightly enhanced activity of the C289A variant compared to the wild-type enzyme may be due to the faster binding of the cysteine substrate in the variant. Indeed, the cluster is more accessible in the variant due to the absence of the fourth cysteine (Cys289) coordinating the cluster. When Cys289 was substituted to alanine, the [4Fe-4S]-cluster was bound to the three conserved cysteines only, with the coordination site on the fourth iron atom being occupied by the hydroxyl group of an ethylene glycol or glycerol molecule coming from the cryo-protectant solution site. Such a coordination suggests that the hydroxyl group of the serine inhibitor or the sulfhydryl group of the cysteine substrate could directly bind to the unique Fe atom of the [4Fe-4S]-cluster of MmCyuA, as illustrated in our structural models of cysteine or serine bound to holo-C289A-MmCyuA (Supplementary Fig. 15E).

Altogether, our mutagenesis and structural results led us to identify the active site amino acids, which are most likely involved in catalysis: the three cysteines that coordinate the [4Fe-4S] cluster and the cysteine/glutamate dyad abstracting the Cα proton of the substrate. Interestingly, the mapping of these residues onto a tree of the complete protein family (containing the methyl-accepting chemotaxis proteins, CyuA and SdaA/B enzymes; Supplementary Fig. 1), show that 1) none of these residues are conserved within methyl-accepting chemotaxis proteins, 2) the three cluster-coordinating cysteines are common to both CyuA and SdaA/B enzymes, which are [4Fe-4S]-dependent, 3) the cysteine/glutamate residues involved in proton abstraction of the cysteine substrate are unique to CyuA enzymes.

The identification of the catalytic residues allows us to propose a full catalytic mechanism for [4Fe-4S]-dependent cysteine desulfidases (Fig. 6), several steps of which had been suggested earlier[20]. First, it is presumed that, upon substrate binding, Cys289, initially bound to the [4Fe-4S]-cluster of MmCyuA, would be labile enough to move away from the cluster (Fig. 6, step 1) and that, in the catalytically active state, the [4Fe-4S] cluster would be bound to the protein by the three conserved cysteines only, as observed in thiouracil desulfidase[55] and several tRNA thiolation enzymes[40], with the coordination sphere being completed by the cysteine substrate (Fig. 6, step 2). In this configuration, the Lewis acidity of the [4Fe-4S] cluster[56] helps polarizing and then cleaving the C-S bond of the cluster-bound cysteine substrate. Next, we suggest here that the highly conserved Glu25, and not Cys23[20], plays the role as the base that deprotonates the Cα carbon of the cysteine substrate (Fig. 6, step 3) because its carboxylate group lies ~5 Å away from the Cα atom of cysteine in the model of the holo-C289A-MmCyuA/cysteine complex (Supplementary Fig. 15E). Yet, Cys23 of MmCyuA is also a catalytically essential residue. In MjCyuA, the equivalent residue, Cys25, was the site of alkylation with N-ethylmaleimide, and its substitution to alanine reduced the activity to 0.4%[20]. In the crystal structures of MmCyuA, Cys23 is located near the anion binding site, rather than at proximity of the cluster, and it is less conserved than Glu25 (Supplementary Fig. 10). Thus, we propose that, as reported previously for the human DJ-1 protein[57], the presence of Arg223, in the environment of Cys23, could lower its pKₐ so that its thiolate group could accept a proton from Glu25 and stabilize the carboxylate form of the latter. After the Glu25-mediated deprotonation step, a so-called [4Fe-5S] species is formed, with a hydro-sulfide bound to the unique Fe atom of the cluster (Fig. 6, step 3). Such a [4Fe-5S] cluster has previously been observed during the desulfidation of 4-thiouracil by the [4Fe-4S]-dependent thiouracil desulfidase TudS[55,58,59] and proposed for other [4Fe-4S]-dependent enzymes, such as tRNA thiolation enzymes[40,60]. The hydrosulfide ligand can then be released for fueling sulfuration reactions (Fig. 6, steps 6 and 7), such as thionucleoside biosynthesis in tRNAs, as shown here (Supplementary Fig. 9). By facilitating sulfur transfer from L-cysteine to tRNA sulfurating enzymes, [4Fe-4S]-dependent CyuA enzymes perform a function similar to that of PLP-dependent CSDs in anaerobic organisms lacking this enzyme. The reaction catalyzed by CyuA also produces 2-aminoacrylate, which tautomerizes into 2-iminopropionic acid (Fig. 6, step 4), with hydrolysis of the latter giving rise to the final products ammonia and pyruvate (Fig. 6, step 5).

Such a function of a [4Fe-4S] cluster acting as a Lewis acid[56] was first proposed for aconitase[61] and later for [4Fe-4S]-dependent LSDs[34], which produce two of the same final products as CyuA, ammonia and pyruvate. [4Fe-4S]-dependent LSDs are the closest structural homologues of CyuA (Fig. 5), in agreement with the close phylogenetic relationship between the two enzyme families (Supplementary Fig. 1). This structural similarity is likely due to the close chemistry of the two reactions, cysteine desulfidation being analogous to serine dehydration, with the removal of $H_2S$ by CyuA and $H_2O$ by LSD, using a [4Fe-4S] cluster as a cofactor. The LpLSD crystal structure showed that all four iron atoms of the cluster are coordinated with protein cysteinyl residues: the three conserved cysteines and the C-terminal cysteine (Cys458), which acted as the fourth ligand and exhibited the same orientation as expected for the L-serine substrate (Fig. 5B)[34]. Accordingly, cysteine behaved as a competitive inhibitor of LpLSD with a $K_i$ of 55 $\mu$M[34], which is in the same range as the inhibition constant of serine that we determined for MmCyuA ($K_i$ of 15 $\mu$M). His61 in LpLSD was suggested to serve as the catalytic base required for abstracting the C$\alpha$ proton of the serine substrate since its substitution to alanine resulted in a complete loss of catalytic activity[34]. Its position, similar to that of Glu25 in MmCyuA, relative to the cluster, supports our assignment of Glu25 as the catalytic base in MmCyuA.

As discussed previously, CyuA appears to be important for providing sulfide (and likely selenide) for the synthesis of various sulfurated/selenidated biological compounds, which would explain its important in vivo function, shown here in the case of *M. maripaludis*. Yet, CyuA could play several other roles in sulfur metabolism, depending on the organism.

First, CyuA could contribute to the regulation of the intracellular concentration of cysteine via its capacity to decompose cysteine catalytically through its cysteine desulfidase activity, as shown in some bacteria[23,26,62]. In this function, CyuA would function more as a cysteine-degrading enzyme than as a sulfide-generating enzyme. Consistently, MmCyuA provides *M. maripaludis* with a protective mechanism from an excess of cysteine when grown under the natural conditions of high sulfide concentration (Fig. 2B). Remarkably, our data also suggest that CyuA provides a mechanism for sustaining growth of microorganisms when cysteine is the sole sulfur source and sulfide is absent. Cysteine was found previously to be unable to support normal growth of *M. maripaludis* as a sole sulfur source[63]. When repeating this experiment, we could observe here very slow growth, whereas Payne et al. recently reported rapid growth[64], which may come from strain differences. In contrast, selective deletion of the *cyuA* gene was shown to prevent any cysteine-dependent growth (Fig. 2C).

Second, CyuA may play a crucial role in regulating intracellular sulfide concentrations. In methanoacocci, CyuA could help cells tolerate high sulfide levels. Notably, *Methanococcales* distinguish themselves from many other methanogens by thriving in environments with elevated sulfide concentration[31], a trait that may be linked to the presence of CyuA, encoded by a gene that is rare among archaea outside this clade (Supplementary Fig. 2B). Moreover, the function of CyuA is likely more significant in *Methanococcales*, which uniquely lack a CSD, compared to other CyuA-containing prokaryotes (Supplementary Fig. 3). Consistent with this, our data indicate that MmCyuA is important under sulfide-rich conditions because the $\Delta mmp1468$ deletion mutant exhibited poor growth, when sulfide served as the sole sulfur source in a cysteine-free medium. In this context, MmCyuA may not act as a cysteine desulfidase but rather as a sulfide transferase, facilitating the transfer of sulfide to enzyme partners involved in sulfur incorporation reactions, potentially via a [4Fe-5S] cluster (Fig. 6, steps 6 and 7). Yet, this interesting hypothesis regarding sulfur assimilation via CyuA warrants further investigation.

## Material and methods
### Phylogenetic analysis
We used the protein database assembled in Garcia et al.[9] that covers the diversity of prokaryotes. Briefly, the genomes were downloaded from NCBI and several rounds of sampling were performed, based on taxonomic redundancy, whole genome comparison, taxonomic clustering and similarity-based clustering using RpoB as a marker. The representative genomes were selected using quality/reference indicators: the number of proteins annotated as reviewed in Uniprot, the NCBI completeness status, the NCBI representative status, and the availability of annotation files. The final database encompasses 9596 bacterial proteomes and 1268 archaeal proteomes. We also used the same taxonomic subsampling of 551 proteomes used in Garcia et al.[9]. It encompasses a number of proteomes per phylum/major clade of the same order and thus corresponds to a homogeneous sampling.

We identified the homologues of CyuA, using iterative BLASTP v2.8.1 +[65] and CyuA sequence of *Escherichia coli* as a seed (YP_026202.1), with an e-value threshold of 1e$^{-4}$. Then, we built aligned sequences using MAFFT v7.419[66] (auto option) and built an HMM profile using HMMER v3.2.1[67] (hmmbuild). We performed an HMM search using this profile on the database and filtered using 0.01 as an e-value threshold. Sequences were realigned and the aligment was manually refined. A preliminary phylogeny of all CyuA homologues was inferred using FASTTREE[68] v2.1.10 (LG + G4) and was used to delineate the CyuA subfamily and remove the paralogues (LSD and methyl accepting chemotaxis protein), based on phylogenetic distance with the other families, the number of copies per genome, the difference of length of sequences, annotations, and specific genomic contexts. The homologues of aerobic markers were identified using the same approach (seeds: *E. coli* KatG catalase NP_418377.1, *E. coli* CyoB Cytochrome bo3 ubiquinol oxidase subunit I NP_414965.1, *Mycobacterium tuberculosis* DesA NADPH-dependent stearoyl-CoA 9-desaturase CCP46048.1, *E. coli* HemF Oxygen-dependent coproporphyrinogen-III oxidase NP_416931.1, *Klebsiella pneumoniae* HpxO FAD-dependent urate hydroxylase ABR77094.1, *Bacillus subtilis* Nos Nitric oxide synthase oxygenase NP_388644.2, *Streptomyces purpurascens* RdmE Aklavinone 12-hydroxylase AAA83424.1).

The dataset of CyuA and LSD was sampled (taxonomic subsampling of 551 representative proteomes), realigned using MAFFT-linsi and trimmed by BMGE[69] v1.12 (BLOSUM30, -b 1 -w 1 -h 0.95). The phylogeny was inferred using IQTREE v1.6.10[70], using the best suited model according to BIC criteria (LG + R7). 1000 replicates of ultrafast bootstraps were computed to assess the robustness of branches. The tree was rooted using LSD SdaA/B sequences, considering that LSD is more ancient than CyuA since it is probably present in the Last Bacterial Common Ancestor due to its wider distribution and a split between Terrabacteria and Gracilicutes in the phylogeny. A tree of CyuA, CSD, LSD, and methyl accepting chemotaxis proteins was also inferred using the same methodology. The reference phylogenies of Bacteria and Archaea of Garcia et al.[9] were used to map the taxonomic distribution of CyuA. Shortly, the bacterial tree was inferred from a concatenation of RpoB, RpoC and IF2 and the archaeal tree was inferred from a concatenation of 30 ribosomal proteins, and for both, IQTREE was used with mixture model (LG + C60 + PMSF). The figures were generated using iTOL[71]. The ecological niches of organisms were identified using METACAT (Guillaume Borrel, personal communication).

### Deletion of the *cyuA* gene in *M. maripaludis*
The *mmp1468* gene was replaced with the *pac* (puromycin N-acetyltransferase) cassette for puromycin resistance in *M. maripaludis* S2, as described[72], resulting in the construction of the $\Delta mmp1468$::*pac* deletion mutant ($\Delta mmp1468$). Primers containing the *SfiI* restriction sites, regions upstream (or UP) and downstream (or DOWN) of the target *mmp1468* gene were PCR amplified using Phusion® DNA polymerase (NEB) Phusion®, and settings were optimized for A + T rich region amplifications[73,74] (Supplementary Table 3). The RE-Marker-RE component of p5L-R was likewise amplified in parallel using the Q5® DNA Polymerase (NEB). The UP and DOWN were then ligated at the *SfiI* sites into the RE-Marker-RE. The upstream and downstream sequences of the first and second REs are already present in the p5L-R and serve as suitable anchor sites for RE-Marker-RE PCR amplification.

### Growth media and culture conditions

Media were prepared in glassware cleaned with 1 M HCl to ensure removal of contaminating sulfur compounds. Unless specified differently, *M. maripaludis* was cultured in anaerobic formate medium under a gas phase of $N_2$/$CO_2$ (80:20) as previously described[75], except that it was reduced with 3 mM DTT instead of cysteine. The stock solution of 0.5 M L-cysteine was prepared in deionized water, filter sterilized, and stored under $N_2$ gas. For most experiments, cultures of *M. maripaludis* strain S2 and Δ*mmp1468* were diluted to a "low" inoculum size of $10^5$ cells per 5 mL of formate medium to minimize the contribution of possible second site mutations. Since both the wild-type and mutant failed to grow with 10 mM cysteine as the sole sulfur source with a "low" inoculum, a "high" inoculum size of $10^7$ cells per 5 mL of formate medium was used. For these experiments, uninoculated medium was tested for the abiotic formation of sulfide with the methylene blue test[63]. Because the test was inhibited by high concentrations of cysteine and DTT, the cysteine concentration was reduced to 1 mM and DTT was omitted. After 6 days at 37 °C, no sulfide was produced with a limit of detection of 1 μM.

Cultures were then incubated at 37 °C, and five replicates of each growth condition were tested. The OD was determined at 600 nm using a Thermo Scientific™ GENESYS™ 20 Visible spectrophotometer. Uninoculated medium with the same composition served as the control. The initial zero-time absorbance of each inoculated tube was subtracted from subsequently measured absorbances for each tube to control for absorbance differences due to variations in the glassware. Growth rate throughout the time course of growth for each culture was calculated via Eq. 1 below. The lag time was defined as the time from inoculation for an individual culture to reach an OD of 0.1. The cell yield was determined directly from maximum OD in stationary phase.

$$Growth\ Rate = \frac{\ln\left(OD_2 \div OD_1\right)}{T_2 - T_1} \qquad (1)$$

### Heterologous gene overexpression of MmCyuA and its variants

MmCyuA from *M. maripaludis* strain S2 (NC_005791.1) was produced from the synthetic *mmp1468* gene, with codon optimization for expression in *E. coli* by Eurofins and subcloned into pET15b, between the NdeI and BamHI restriction sites, to add a 6-His cleavable tag at the N-terminus, which could be cleaved by the H3C protease. Mutagenesis was performed by Genescript. Competent *E. coli* BL21 (DE3) cells were transformed with the *cyuA*-containing plasmids to overexpress MmCyuA and its variants. A single colony was used to inoculate 200 mL of Luria Broth (LB) medium supplemented with ampicillin (100 μg.mL$^{-1}$). 40 mL of this preculture, grown overnight at 37 °C, was then used to inoculate 4 L of the same medium. Cells were incubated at 37 °C until OD$_{600}$ reached 0.6, and protein expression was induced at 16 °C with 0.5 mM isopropyl-γ-D-thiogalactopyranoside. Cells were incubated for 18 h at 16 °C, collected by centrifugation at 4000 × *g* at 4 °C for 15 min, and stored at −80 °C until use.

### Aerobic purification of MmCyuA and its variants

Cells were resuspended in 25 mM HEPES pH 7.5, 500 mM NaCl, 10% glycerol, containing 2 μg.mL$^{-1}$ RNase A (Fisher), protease inhibitor cocktail (Roche, one tablet per 50 ml), 5 mM β−mercaptoethanol, 1 mM MgCl$_2$, 5 mM imidazole, and disrupted by sonication. Cells debris were removed by centrifugation at 35,000 rpm for 1 h at 4 °C. The supernatant was then purified by Ni-NTA affinity chromatography using 2 × 5 mL HisTrap™ FF columns (Cytiva) in 25 mM HEPES pH 7.5, 200 mM NaCl, 5 mM β−mercaptoethanol using a linear gradient of 5–500 mM imidazole at 1.5 mL.min$^{-1}$ in 100 min. The protein was collected, dialyzed twice against 3 L of 25 mM HEPES pH 7.5, 200 mM NaCl, 5 mM β−mercaptoethanol in the presence of the H3C Protease (25 μg per mg MmCyuA). After concentration with a Vivaspin 20 ultrafiltration membrane (30 kDa cutoff, Sartorius), the protein was further purified at 1 mL.min$^{-1}$ onto a Hiload 16/60 Superdex 200 gel filtration column (Cytiva) equilibrated with 25 mM

HEPES pH 7.5, 200 mM NaCl, 5 mM β−mercaptoethanol, using an ÄKTA system. The 'as-purified' protein was concentrated to 4 mg.ml$^{-1}$ with an Amicon Ultra filter device (30 kDa cutoff), frozen in liquid nitrogen and stored at −80 °C. The purity of MmCyuA was assessed along the purification steps using SDS-PAGE gels, and its concentration determined using the Bradford assay (Biorad)[76] using bovine serum albumin as the standard.

The GST–H3C-protease (a gift from S. Mouilleron) was expressed using pGEX-2T recombinant plasmids. After induction at 25 °C with 0.1 mM IPTG for 20 h, the protein was purified using glutathione–Sepharose chromatography.

### In vitro [Fe-S] cluster reconstitution and anaerobic purification of holo-MmCyuA and its variants

The reconstitution of the [4Fe-4S] cluster and purification of holo-MmCyuA and its variants were performed in a glove box (MBraun) containing less than 0.5 ppm O$_2$. After incubation of as-purified MmCyuA (100 μM) with 10 mM DTT for 15 min, a 5-fold molar excess of ferrous ammonium sulfate and sodium sulfide was added. Cluster reconstitution was followed by the appearance of a band at around 410 nm in the UV-visible spectrum, which was recorded with an XL-100 UVICON spectrophotometer equipped with optical fibers, and incubation was stopped after 2–3 h when OD$_{410}$/OD$_{280}$ > 0.3. After centrifugation for 30 min at 20,000 × *g*, holo-MmCyuA was loaded onto a Superdex 200 Increase 10/300 GL gel filtration column (Cytiva) equilibrated in 25 mM HEPES pH 7.5, 150 mM NaCl. After determination of the Fe and S contents using the Beinert and Fish methods, respectively[77,78], the purified holo-proteins were concentrated around 10 mg.mL$^{-1}$ on a Vivaspin concentrator (30 kDa cutoff, Sartorius). After freezing and storage in liquid nitrogen in sealed tubes, the [4Fe-4S] cluster displayed degradation, as shown by the poor activity and [Fe-S] content of the protein, as well as the [2Fe-2S] state of the cluster observed in Crystal 2. Therefore, for subsequent crystallization and activity tests, the holo-proteins were used immediately after reconstitution assays.

### EPR spectroscopy

EPR measurements were performed on holo-MmCyuA, after aerobic purification and anaerobic cluster reconstitution on a Bruker ELEXSYS-E500 spectrometer X Band, operating at 9.3881 GHZ and equipped with a SHQE cavity and a helium flow cryostat (ESR 900 Oxford Instruments). Spectra were recorded at low temperature in the range of 10 to 40 K, under non-saturating conditions, using microwave powers of 2 and 10 mW, a modulation amplitude of 0.4 mT and a modulation frequency of 100 kHz. The EPR spectra of holo-MmCyuA (200 μM), in 25 mM HEPES pH 7.5, 200 mM NaCl, 5 mM DTT were recorded on the frozen samples before and after reduction with 5 mM DTT for 30 min at room temperature.

### SEC-MALS

SEC-MALS experiments were performed using an HPLC-MALS system (Shimadzu) equipped with light scattering detector (mini-DAWN TREOS, Wyatt Technology), refractive index detector (Optilab T-rEX, Wyatt Technology) and UV detector (SPD-20A, Shimadzu). Apo- or holo-MmCyuA (100 μL at 2 mg.mL$^{-1}$) was injected on a Superdex 200 10/300 GL Increase column (Cytiva) equilibrated in 50 mM Tris-HCl pH 8.5, 300 mM NaCl, 5 mM DTT at a flow rate of 0.5 mL.min$^{-1}$. The molar mass was calculated with the ASTRA 6.1 software (Wyatt Technology) using the Forward Monitor mode to take into account the fact that the holo-protein absorbs light at the laser wavelength[38]. A refractive index increment (dn/dc) value of 0.183 mL.g$^{-1}$ was used.

### Cysteine desulfidase activity test

Apo-MmCyuA, holo-MmCyuA or the holo-C23A, E25A, E25D, E25Q and C289A MmCyuA variants (1 μM) were incubated in the presence of 0.05-5 mM L-cysteine at 37 °C for 2–30 min in 25 mM HEPES pH 7.5, 150 mM NaCl and the reaction was stopped by adding 1/10 (v/v) 1 M HCl. The

formation of the pyruvate product was measured by the absorbance at 340 nm after derivatizing with 2,4-dinitrophenylhydrazine[39]. $Mg^{2+}$, which was included in the activity test of MjCyuA[20], was not necessary for cysteine desulfidase activity. L-seleno-cysteine was prepared by reducing 40 mM L-seleno-cystine (Sigma) with 200 mM DTT in 100 mM Tris-HCl, pH 8.0 for 30 min[79]. L-serine inhibition assays were performed using 0.05–1 mM L-serine and 0.1–1 mM L-cysteine. The pyruvate formation was first recorded for 30 min to determine the linear phase of the product formation versus time curve and measure the initial velocity ($Vi$) from its slope using linear regression. The kinetic parameters, $k_{cat}$ and $K_m$, for L-cysteine and L-selenocysteine, were then determined after stopping the reaction after 2 min using the $Vi$ as a function of substrate concentration plot by nonlinear regression fitting of the Michaelis-Menten equation using the GraphPad Software (Prism version 9.0.0 for Windows, Boston, Massachusetts, USA)[80]. The inhibition by L-serine was analyzed with a Lineweaver–Burk plot, by plotting the reciprocal of initial reaction rate ($1/V_i$) against the reciprocal of substrate concentration (1/L-cysteine) at different concentrations of L-serine (0.05 to 1 mM). The inhibition constant $K_i$ was determined by plotting the slopes of the Lineweaver-Burk plots as a function of L-serine concentration (Dixon plot). The coordinate on the x-axis of the intercept of the slopes of the secondary plot divided by the intercept is equal to $-K_i$.

### Sulfur transfer assays by MmCyuA from L-cysteine to the tRNA thiolation enzymes TtuI and NcsA

MmCyuA (apo or holo) was incubated for 1 h in 25 mM HEPES pH 7.5, 200 mM NaCl at 37 °C with MmTtuI or MmNcsA (apo or holo) and 15 μM tRNA$^{Lys}_{UUU}$ transcript, 1 mM ATP and 2.5 mM $MgCl_2$ in the presence or absence of cysteine (1 mM) or $Na_2S$ (1 mM) as the sulfur source. Preparation of tRNA by T7-RNA polymerase transcription, tRNA digestion with nuclease P1 and alkaline phosphatase and analysis of modified nucleosides using HPLC-coupled tandem mass spectrometry were described in ref. 41.

### Crystallization, X-ray data collection and processing

All crystals of holo-CyuA at 8 mg.mL$^{-1}$, in 25 mM HEPES pH 7.5, 150–200 mM NaCl, were grown anaerobically at 18 °C using the hanging drop method and flash-frozen under anaerobic conditions using liquid propane after cryoprotection with 20-25% glycerol. Crystals of holo-CyuA alone ($0.1 \times 0.15 \times 0.1$ mm) were obtained by mixing 2 μL of protein with 1 μL of a 500 μL reservoir solution containing 13% PEG 4000, 0.1 M NaCl, 0.1 M tri-sodium citrate dihydrate pH 5.5, 0.1 M $LiSO_4$, 10% glycerol. Crystals of holo-CyuA with MES were grown by mixing 1 μL of protein and 10 mM L-serine with 1 μL of a 500 μL reservoir solution containing 14% PEG 4000, 0.1 M MES pH 6.5, 0.1 M NaCl, 0.1 M $MgCl_2$. Crystals of holo-MmC289A-CyuA were grown by mixing 1 μL of protein at 7 mg.mL$^{-1}$ with 1 μL of 500 μL solution containing 12% PEG 4000, 0.1 M NaCl, 0.1 M tri-sodium citrate dihydrate pH 5.5. Some of these crystals were cryoprotected in the same solution containing 25% ethylene glycol, whereas others were soaked in the same solution containing 100 mM serine, then cryoprotected with 25% glycerol. X-ray data were collected on a single crystal at 100 K at the SOLEIL synchrotron (Saint Aubin, France) on the PX1 and PX2 beamlines. Native and iron anomalous (7.130 keV) diffraction data were collected for all crystals. Data were indexed, processed and scaled with the autoPROC pipeline[81] and the output from STARANISO[82] was used as the mtz reflection file.

The Fe substructure of the holo-MmCyuA-MES structure was solved using SHELXC and SHELXD[83] using data collected at 7.130 keV. Initial phases were calculated with Phaser-EP[84] using the previously obtained substructure as input. Phases were improved with 5 cycles of density modification with PARROT[85] (FOM = 0.69). The resulting 4 Å resolution map was used to build an initial model in COOT[86] consisting of manually placed secondary structure elements using the structure of LpLSD (PDB code 4RQO) as a guide. This model was used as input in CRANK2[43] in the MR-SAD mode, which built 787 residues with a FOM of 0.93 after 6 cycles of model building ($R_{work}$= 0.25, $R_{free}$ = 0.32). The final model was obtained by alternating building in COOT[86] and refinement in BUSTER[87].

The holo-MmCyuA structure was solved using the AlphaFold prediction of *M. jannaschii* CyuA (ID: AF-Q58431-F1; 52.8% sequence identity, Supplementary Fig. 10A) by molecular replacement using *PHENIX-Phaser*[88]. All models were refined by alternating manual building in COOT[86] and refinement in *phenix-refine*[89] or *autoBUSTER*[87] using NCS restraints and two TLS groups per chain.

The models of the complexes of holo-MmC289A with cysteine and serine were built manually in COOT[86] by replacing the glycerol ligand by cysteine/serine and positioning the SH group and OH group of cysteine/serine 2.4 and 2.3 Å away from the cluster, respectively.

### Statistics and reproducibility

All data shown are mean values of at least three independent biological replicates, with error bars denoting the standard deviation, unless indicated otherwise in the figure caption. Statistical analyses for cell growth data were performed using the two-way t-test assuming unequal variance. The p-values for the two-way t-test are reported in Supplementary Table 2. When the software used is not specified, calculations were performed in Microsoft Office Excel with built-in shortcuts to process information such as the mean and standard deviation.

### Reporting summary

Further information on research design is available in the Nature Portfolio Reporting Summary linked to this article.

### Data availability

All data generated or analyzed during this study are included in the text and figures of this article and its Supplementary Materials. The sequence alignments and phylogenetic trees are provided as Supplementary Data 1–8 and the source data for all figures (in the main article and the Supplementary Materials file) are provided as Supplementary Data 9–12 (see the list in the Supplementary Materials file). All other data are available from the corresponding author upon reasonable request. The atomic coordinates and structure factors of all crystal structures were deposited in the Protein Data Bank (PDB codes 9FYI, 9FSL, 9HE2 and 9HE0).

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

## Acknowledgements

We thank SOLEIL for provision of synchrotron radiation facilities, Pierre Legrand and Martin Savko for assistance in using beamlines PROXIMA-1 and 2, the French EPR CNRS Facility, Infrastructure de Recherche Renard (IR 3443) and Frédéric Barras for discussions about phylogeny and evolution. This work has benefited from the facilities and expertise of the Macromolecular Interaction Platform of I2BC and was supported by the French State Program 'Investissements d'Avenir' (Grants "LABEX DYNAMO", ANR-11-LABX-0011 and "Integrative Biology of Emerging Infectious Diseases", ANR-10-LABX-62-IBEID), the French National Research Agency (grant "sulfo-tRNA", ANR-22-CE44-0012), the European Union's Horizon 2020 research and innovation programme under the Marie Skłodowska-Curie grant agreement No 101034407, the Centre National de la Recherche Scientifique, the Institut Pasteur and the Division of Chemical Sciences, Geosciences, and Biosciences, Office of Basic Energy Sciences of the U.S. Department of Energy through Grant DE-SC0018028.

## Author contributions

Conceptualization: W.B.W., B.G.-P. Methodology: S.G., P.Z., E.B.S., N.H. L.P., P.S.G., N.T., C.V., J.-L.R., B.F. Investigation: S.G., P.Z., E.B.S., N.H., L.P., P.S.G., T.A., N.T., O.B., C.V., J.-L.R., B.F. Visualization: S.G., P.Z., E.B.S., N.H., L.P., P.S.G., N.T., C.V., B.G.-P Supervision : W.B.W., B.G.-P. Writing—original draft: P.S.G., W.B.W., B.G.-P. Writing—review and editing: S.G., L.P., P.S.G., W.B.W., M.F., B.G.-P.

## Competing interests

The authors declare no competing interests.

## Additional information

[1]Laboratoire de Chimie des Processus Biologiques, UMR 8229 CNRS, Collège de France, Sorbonne Université, Paris, France. [2]Department of Microbiology, University of Georgia, Athens, GA, USA. [3]Stress Adaptation and Metabolism in Enterobacteria and Evolutionary Biology of the Microbial Cell Units, Institut Pasteur, Université Paris Cité, Paris, France. [4]IR CNRS Renard, Chimie-ParisTech, Paris, France. [5]Institute for Integrative Biology of the Cell (I2BC), CEA, CNRS, Université Paris-Saclay, Gif-sur-Yvette, France. [6]University of Grenoble Alpes, CEA, CNRS, IRIG, Grenoble, France. [7]Present address: Laboratoire de Chimie Bactérienne, Institut de Microbiologie de la Méditerranée, Institut Microbiologie Bioénergies et Biotechnologie, Aix-Marseille Université, Centre National de la Recherche Scientifique UMR 7283, Marseille, France. [8]Present address: Laboratoire de Chimie des Polymères Organiques, UMR 5629, CNRS, Université de Bordeaux, 33600 Pessac, France, UMR 5234 CNRS-University of Bordeaux, SFR TransBioMed, Bordeaux, France. [9]These authors contributed equally: Sylvain Gervason, Paolo Zecchin, Elliot B. Shelton. ✉e-mail: beatrice.golinelli@college-de-france.fr

