## [Transparent Peer Review file · Communications Biology]

Evolution, structure and function of L-cysteine desulfidase, an enzyme involved in sulfur metabolism in the methanogenic archeon *Methanococcus maripaludis*

Corresponding Author: Dr Béatrice Golinelli-Pimpaneau

Version 0:

Reviewer comments:

Reviewer #1

(Remarks to the Author)

The authors describe the species distribution of genes encoding L-Cys desulfidases and characterize the biochemistry, spectroscopy, and structure of the enzyme from *Methanococcus maripaludis*. They provide compelling evidence that the [4Fe-4S] cluster in the protein facilitates sulfur transfer from L-Cys or sulfide to tRNA sulfuration enzymes via a [4Fe-5S] intermediate. The studies are carefully performed and clearly described. The following concerns are relatively minor.

Line 44: A mutant is a cell containing a mutation, not a protein with a substituted residue. The word "mutant" should be replaced by "variant" here and elsewhere (lines 296, 341, 342, 627).

Line 62: A persulfide has two sulfur atoms, so the text should read "the sulfane sulfur atom".

Line 75: change "sulfur" to "sulfide".

Line 79: In Table S1, should "methylreductase" be "methyl-coenzyme M reductase"?

Line 113: Figure S1 is unreadable. The alignment file is provided separately, so this figure either should be deleted or altered to provide useful information. The homology between CyuA and methyl-accepting chemotaxis proteins raises the question of whether S-adenosylmethionine inhibits the desulfurase activity like L-serine or if it serves as a substrate for methylsulfide transfer. Were these points tested?

Line 114: The two columns of numbers in Figure S2 would benefit by using a larger font size.

Line 128: The six markers chosen for identifying aerobic microorganisms in Figure S4 seem somewhat arbitrary. For example, how widely distributed is aklavinone 12-hydroxylase and the other enzymes chosen as markers? Are similar patterns observed if using cytochrome or quinol oxidases? In my opinion, the conclusions from this analysis represent the least convincing feature of this manuscript.

Line 145: The growth curves presented in Figure 1 would be more useful by showing the y-axis as $\ln(\text{OD})$ so that the timing of exponential growth is more readily visualized.

Lines 178-181 can be rewritten to improve clarity.

Line 212: Should the infinity symbol subscript be a perpendicular symbol?

Line 243: Table 1 indicates the structures are resolved at rather poor resolution and the precise orientation of the [FeS] cluster and ligand (line 250, Figure 3) is uncertain, but the text is careful not to overinterpret the results, so this is OK.

Line 247: cite the PDB ID of LpLSD.

Line 257-260: This sentence could be rewritten for better readability.

Lines 267 and 309: A comment would be appropriate to explain why the occupancy of [4Fe-4S] cluster is 65% for holo-MmCyuA, but it is fully occupied for holo-C289A-MmCyuA.

Line 289: Any explanation for why Holo-C289A-MmCyuA shows greater activity than holo-MmCyuA (Fig.2A)?

Line 295: DNA is mutated, not proteins. Change "mutated to alanine" either to "mutated the codon for alanine" or "substituted with alanine". The same issue occurs in lines 340, 405, 426.

Line 296: In Figure S14, change "mutants" to "variants". Consider renumbering Table S4 to Table S3 to retain sequential numbering. Also, change "mutants" to "variants" in the title of the table.

Line 370: "CyuA can be used as a carbon/energy source" might be true if the protein were degraded. The authors mean that "CyuA can metabolize L-cysteine as a carbon/energy source".

Line 386: Define MjCyuA.

Line 414: Figure 5 suggests the protonated thiol group of substrate binds to the unique Fe of the [4Fe-4S] cluster and that the cluster-bound sulfide is protonated. No experimental evidence was reported to address the protonation states. The authors

are encouraged to indicate the deprotonated species may alternatively occur in both cases.

Line 513: Fix the typo ("probalbly"). Also, this paragraph has nine uses of "has been" or "have been" that could be simplified to "was" or "were". The same issue is found elsewhere throughout the text.

Line 585: replace "BSA" with "bovine serum albumin".

Line 609: use uppercase for 40K.

Reviewer #2

(Remarks to the Author)

I co-reviewed this manuscript with one of the reviewers who provided the listed reports. This is part of the Communications Biology initiative to facilitate training in peer review and to provide appropriate recognition for Early Career Researchers who co-review manuscripts.

Reviewer #3

(Remarks to the Author)

The article "Evolution, structure and function of L-cysteine desulfidase, an enzyme involved in sulfur metabolism in methanogenic archaea" authored by Sylvain Gervason, Paolo Zecchin, Elliot B. Shelton, et al. reports the structure and function of a protein called CyuA (E. coli nomenclature). Cysteine desulfidases are very interesting proteins that are certainly worth to have a closer look at. From this point of view and with respect to the quality of this trans-disciplinary study combining evolutionary biology, extraordinary structural biology and biochemistry, this paper must certainly be published as soon as possible.

However, some minor and one major point should be addressed prior publication:

Minor points:

1) lines 141-147: The authors say that the archaeon *M. maripaludis* is usually cultivated in 3-5 mM Na₂S but still cultivated them in 2mM calling this "optimal" concentrations. Can the authors explain why they don't use 3-5 mM?

2) lines 154: *M. maripaludis* (MmCyuA): the authors may put the proteins name not in brackets. This may lead to misunderstanding.

3) lines 154-155: it is misleading that both, high and low levels, are described as "abundant".

4) line 159: can the authors give an example where we actually do find high concentrations in the environment?

5) lines 175-176: this strategy should have also been tested in the initial growth tests. If the authors have this piece of data it would be nice to include it to the manuscript.

6) line 178: "the abiotic formation of sulfide from cysteine was not detected". How did the authors measure that, is there data to show?

7) line 195: "after treatment with dithionite and EDTA to remove the residual cluster (Figure S5C, 7 dotted line), SEC-MALS analysis of 'apo-MmCyuA' thus obtained indicated that it was mainly a dimer (Figure S5D)." Please rephrase.

8) lines 226-228: the study would greatly profit from including affinity measurements with respect to the different amino acids using e.g. MST (microscale thermophoresis).

9) lines 257-258: are there inter-domain connections that allow "communication" between the two subunits such as found in cysteine desulfurases? The paper would benefit from a closer look into the structure.

10) line 281: the authors state that the two molecules in the asymmetric unit were only "similar". Are there notable differences?

11) line 282: why is there a 2Fe/2S found when the authors co-crystallize the protein with an inhibitor? Does this have any functional meaning?

12) lines 277 and following: what is the benefit of the paragraph "The crystal structure of the [2Fe-2S]-MmCyuA in complex with MES confirms the location of an anion binding site"? It may be worth to combine this paragraph with the following one and shorten it.

13) line 303: the authors state "several loops". Please, be more precise. There is no useful info in that statement, yet.

14) line 300: It would be interesting to soak the WT enzyme with the molecules found in the structure of the mutant enzyme as well as the substrate and the product. Is there any reason the authors did not do this?

15) line 309: "Fe edge" do the authors mean "Fe K edge"?

16) line 314: the presence of hydrogen bonds is rather a suggestion at a resolution of 3.2 Å.

17) line 345: "The C-terminal domain of MmCyuA displays strong structural similarity with the catalytic domain of serine dehydratases." What is the benefit of this data for the manuscript? Does it need a separate paragraph? It may be incorporated as only one sentence in the paragraph "Cys23 and Glu25 of MmCyuA are critical for the cysteine desulfidase activity"

18) line 385: "first homogeneous anaerobic preparation": this is not fully correct! It was prepared aerobically and then reconstituted anaerobically. Please correct.

19) lines 436/437: rephrasing needed.

Major point:

In lines 188-189 the authors state that they purified the protein aerobically and in line 391 they explain that under such conditions Fe/S clusters may be degraded. Why do the authors not purify the protein anaerobically? They have shown that they have the expertise to do so.

Notably, the degradation on one hand is very common under aerobic conditions and on the other hand the reconstitution of proteins with Fe/S clusters does not necessarily yield the physiological type of Fe/S cluster. In fact, very often during chemical reconstitutions non-physiological 4Fe/4S clusters are incorporated into proteins that in vivo do not carry such cluster types. It is therefore strongly recommended to purify the protein, both WT and mutants, anaerobically and test for physiological cluster content.

For future projects and with respect to physiology, of course an overexpression and purification from an archaeal host would be the gold standard, since *E. coli* may even in vivo equip enzymes with the wrong type of cluster. However, setting up such a procedure may take some time and hence this is a suggestion for the next study.

Reviewer #4

(Remarks to the Author)

Gervason et al characterize the taxonomic distribution, evolution, biochemical function, physiological function, and structure/function of L cysteine desulfidase (CyuA). They show that CyuA was unlikely to be a property of LUCA and rather likely emerged in Bacteria, with LGT to Archaea. Heterologous expression of CyuA from MmS2 shows the enzyme catalyzes the decomposition of cysteine to sulfide/ammonia/pyruvate, suggesting a potential function in liberating S for cellular biosynthesis. This is important, as this helps to explain several phenomena in sulfur metabolism of past studies of MmS2. Through deletion studies, the authors then show that this enzyme plays a key role in sulfur metabolism in MmS2, in particular under low sulfide growth conditions. It is suggested that CyuA assists in providing protection against cysteine toxicity. They obtained a structure and showed that the 4Fe4S cluster is required for activity (I will not comment on the structural or protein mutation aspects of this work, as I am not an expert in these areas). Overall, the dataset is comprehensive and fills a knowledge gap in our understanding of S metabolism in anaerobes, in particular MmS2. Overall, I enjoyed reading this manuscript and have a few suggestions to further improve it and to clarify several important points in preparation for readers of *Comm. Biol.*

General Comments:

The abstract would benefit by including some statement regarding the broader relevance/significance of CyuA to the microbiology field. Perhaps including information on the taxonomic distribution among anaerobes and/or the evolution of the protein – is it recent if it was not a property of LUCA? Does it help explain the lack of cysteine desulfurase homologs in some Methanococcales. As it stands, a reader of the abstract may not realize the broader implications of the work and rather may think this is a study on a peripheral protein (it is not but the worry is there).

I appreciated the authors of this largely biochemical study using phylogenetic/evolutionary approaches to make their work of broader importance, yet I was a bit underwhelmed with the treatment. More insight could be gleaned from the evolutionary analysis, including specific mention of where Methanococcales obtained CyuA homologs from. There is so much more hiding in that phylogeny in the SI that could be used to further the relevance of this work. I would even go so far as to suggest that the phylogeny (perhaps a collapsed version) should be included as primary figure in the paper, with combining other figures in the paper to keep within *Comm. Biol.* requirements.

The authors often use extremely high concentrations of substrates in their assays (e.g., 10 mM cysteine) and then use the results of such assays to suggest ecological/physiological roles for CyuA in host organisms. It is hard for me to imagine an environment with concentrations of cysteine that approach the levels used herein and thus suggest such arguments be tempered. Further, the authors should consider whether the high sulfide numbers are accurate or if those studies they cite (>10 mM sulfide in salt marsh sediments) might be attributable to sulfide complexed with metals that would render the sulfide either non-toxic or not available. In other words, if there is really 10 mM of free sulfide, then the cells have major problems as

nearly all metals will be precipitated as sulfide minerals. Alternatively, a fraction of what was called “free sulfide” may be acid dissociable soluble metal sulfide clusters that present as free sulfide in typical methylene blue assays (add acid to dissociate metal sulfide clusters before the assay).

Specific comments:

Line 71: MmS2 has been shown previously to grow on cysteine alone or in combination with thiosulfate (see doi: 10.1038/s42003-024-07049-w). This does not take away from the current paper but it should be mentioned in the paper. Also, somewhere there is a study that showed that Methanococcales can generate sulfide from cysteine despite lacking cysteine desulfurase homologs. This is the paper that comes to mind: doi: 10.1074/jbc.M110.152447.

Line 97: Several members of the Methanococcales do encode SufS (please see Table 1 in doi: 10.1128/JB.00117-2). MmS2, as the authors correctly note, does not.

Line 108: A quick read did not convince me that reference 26 actually made this claim and I would be surprised if this were true (which the authors then show to be the case later in their report). I would refer to Weiss 2016 (doi: 10.1038/nmicrobiol.2016.116) to see if they also pinned CyuA to LUCA (I do not believe they did) and use this work. This would also help clear up the confusion that resulted when I read this sentence and then read later that CyuA was not in LUCA.

Line 124: What was the evolutionary path of CyuA among Archaea? How did Methanococcales end up with this protein? Additional detail would be appreciated here.

Lines 144-145: Sulfide is toxic to MmS2 and cells actually prefer to grow with sulfide that is complexed with other metals such as Fe. It is also true that sulfide results in metal limitation for cells. Please see Fig. 1 in doi: 10.1038/s42003-023-05163-9. Further, cells grown with cysteine + thiosulfate grew nearly as well as those provided with sulfide (2 mM) indicating that exogenous sulfide is not necessary for optimal growth (see again Payne 2024). I suggest that the sentence be rephrased for accuracy.

Lines 161-168: These concentrations of cysteine are not terribly relevant from a cytotoxicity standpoint (where would MmS2 encounter such high concentrations?). I wonder if the effect is attributable to such high concentrations of cysteine making it difficult for the cells to obtain the metals that they need (complexes the metals). Alternatively, is it possible that the role of cysteine in these results is to decrease the ORP of the medium or to scrub out O₂ when insufficient sulfide is available to do so?

Line 495: This is a quite high e value and it is not clear how this value was settled on. I wonder if the authors, now that they have identified active site residues in CyuA, can return to their alignments and show retroactively that the previously identified CyuA homologs are likely to be these proteins? This would also provide a mechanism to validate (and ensure readers) that this e value is appropriate. And it would be a neat way to show how useful the combined biochemical/phylogenetic approach used in this study is and would provide a model for future studies of this type.

Congratulations on a nice study: Eric Boyd.

Version 1:

Reviewer comments:

Reviewer #1

(Remarks to the Author)

The authors have adequately addressed the minor concerns I indicated in the original manuscript. Nice work!

Reviewer #2

(Remarks to the Author)

I co-reviewed this manuscript with one of the reviewers who provided the listed reports. This is part of the Communications Biology initiative to facilitate training in peer review and to provide appropriate recognition for Early Career Researchers who co-review manuscripts.

Reviewer #3

(Remarks to the Author)

I am very happy with the answers given and changes made by the authors.

However, I strongly disagree that anaerobic MST was not possible. It was shown several times that it is well possible. E.g. refer to the works of Roland Lill. In fact it very simple since oxygen does not diffuse rapidly inside the capillary.

<https://resources.nanotempertech.com/application-notes/anaerobic-microscale-thermophoresis-reveals-the-redox-dependency-of-ferredoxin-in-mitochondrial-fe-s-biogenesis>

Since the exact affinity, however, does not improve the manuscript significantly, I think the manuscript should now be published in its present form as soon as possible.

Reviewer #4

(Remarks to the Author)

In general, the authors have adequately addressed my concern with the following exception(s): In my previous comment, I indicated the existence of "...a study that showed that Methanococcales can generate sulfide from cysteine despite lacking cysteine desulfurase homologs. This is the paper that comes to mind: doi: 10.1074/jbc.M110.152447". The reviewers state the following in response:

The authors response is "The reviewer also misread the paper Liu et al. 2010 (doi: 10.1074/jbc.M110.152447). Cysteine was not shown to replace sulfide as a sulfur source in that work. No change was made."

This reviewer did not misread the paper. Rather, the authors misread the paper and my comment. As stated in Liu et al., 2020: "Although cysteine desulfurase homologs have not been identified in the *M. maripaludis* genome, cysteine desulfurase activity was detected in cell-free extracts." This suggests an alternative mechanism (not canonical cysteine desulfurase) to metabolize cysteine. The authors should adjust their description of this finding where necessary throughout the paper.

Secondly, if the authors seek to perpetuate false claims of the presence of CyuA in the ancestor of Bacteria/Archaea (e.g., LUCA) based on a weak study, it is their prerogative to do so. However, it would be cleaner to state that it was not a property of LUCA based on the more robust, recent study of Weiss and leave it at that. Especially since the author's own analyses (CyuA in methanococcales via HGT from bacteria) concur with Weiss.

Version 2:

Reviewer comments:

Reviewer #4

(Remarks to the Author)

16/07/25

Reviewers' comments with responses

Reviewer #1 (Remarks to the Author):

The authors describe the species distribution of genes encoding L-Cys desulfidases and characterize the biochemistry, spectroscopy, and structure of the enzyme from *Methanococcus maripaludis*. They provide compelling evidence that the [4Fe-4S] cluster in the protein facilitates sulfur transfer from L-Cys or sulfide to tRNA sulfuration enzymes via a [4Fe-5S] intermediate. The studies are carefully performed and clearly described. The following concerns are relatively minor.

Line 44: A mutant is a cell containing a mutation, not a protein with a substituted residue. The word “mutant” should be replaced by “variant” here and elsewhere (lines 296, 341, 342, 627).

corrected

Line 62: A persulfide has two sulfur atoms, so the text should read “the sulfane sulfur atom”.

corrected

Line 75: change “sulfur” to “sulfide”.

corrected

Line 79: In Table S1, should “methylreductase” be “methyl-coenzyme M reductase”?

The enzyme is now named “methyl-Coenzyme M methylreductase”.

Line 113: Figure S1 is unreadable. The alignment file is provided separately, so this figure either should be deleted or altered to provide useful information. The homology between CyuA and methyl-accepting chemotaxis proteins raises the question of whether S-adenosylmethionine inhibits the desulfurase activity like L-serine or if it serves as a substrate for methylsulfide transfer. Were these points tested?

Figure S1 has been removed and the text now refers only to the online Figure instead.

We did not check whether S-adenosyl-L-methionine is an inhibitor or a substrate of CyuA. However, S-methylcysteine was previously shown not to be a substrate of MjCyuA (Tchong et al., Biochemistry, 2005) so that methylsulfide transfer is highly unlikely.

Line 114: The two columns of numbers in Figure S2 would benefit by using a larger font size.

We used a larger font size for the two columns of numbers in Figure S2.

Line 128: The six markers chosen for identifying aerobic microorganisms in Figure S4 seem somewhat arbitrary. For example, how widely distributed is aklavinone 12-hydroxylase and the other enzymes chosen as markers? Are similar patterns observed if using cytochrome or quinol oxidases? In my opinion, the conclusions from this analysis represent the least convincing feature of this manuscript.

We chose our selection of markers based on a study of Jablonska and Tawfik, 2019 (Ref 34). Briefly, the authors showed a correlation between the aerobic phenotype and the number of oxygen-utilizing enzymes. These enzymes that are specific to aerobes were mostly involved in secondary metabolism and five of them were identified as the best discriminators between aerobes and anaerobes. We thus selected these markers. To potentially increase the sensitivity of detection of aerobic organisms, we already used an additional marker, the more widespread catalase. Following the suggestion of the reviewer, we now also include the cytochrome bo₃ ubiquinol oxidase subunit I CyoB in the dataset, which increased the coverage of aerobic organisms (see Figure below).

CAT
CycB
DnaA
Hmf
HipX
NoSI
RdmE

CAT
CycB
DnaA
Hmf
HipX
NoSI
RdmE

- Izermarchaia
- Thalassarchaea
- Pontarchaea
- SG8-5
- Methanomassiliococcales
- Thermoplasmatales
- Acidiprofundales
- Archaeoglobales
- Halobacteria
- Methanomonarchaea
- Methanophagales
- Syntrophoarchaeales
- Methanosarcinales
- Methanomicrobiales
- Methanocellales
- Hydrothermarchaea
- Methanococcales
- Methanobacteriales
- Methanopyrales
- Hadesarchaea
- Persephonarchaea
- Thermococcales
- Theilonarchaeia
- Methanofastidiosia
- Altarchaeales
- Diapherotrites
- Micrarchaeota
- Nanoarchaeota
- Aeomarchaeota
- Huberarchaea
- Parvarchaeota
- Nanoarchaeota
- Woesearchaeota
- Pacesarchaeota
- Korarchaeota
- Helicoflexarchaeota
- LoRiarchaeota
- Oviarchaeota
- Thovarchaeota
- Verstraenarchaeota
- Marsarchaeota
- Crenarchaeota
- Bethyarchaeota
- Thaumarchaeota
- Algararchaeota
- Geothermarchaeota

We thank the reviewer for this suggestion as it further exacerbated the trend that we already observed, with a stronger difference between CyuA-containing genomes and other genomes regarding the number of oxygen-utilizing markers (see New Figure S4).

Line 145: The growth curves presented in Figure 1 would be more useful by showing the y-axis as $\ln(\text{OD})$ so that the timing of exponential growth is more readily visualized. *While we realize that log plots are a common practice, growth of methanogens is usually substrate limited and linear plots are the norm for this field. Linear plots have the advantages of showing yields and lag phases more clearly, which were critical to the interpretation of these results. No change was made.*

Lines 178-181 can be rewritten to improve clarity.

Reviewer 3 had a comment on this same section, which was rewritten to clarify. More detail on the test for abiotic sulfide formation was included in the methods on lines 542-545. Both sections were rewritten.

Lines 178-181 were rewritten as: "In this medium, the abiotic formation of sulfide from 1 mM cysteine was below the limit of detection. When the S2 strain was grown in the absence of sulfide, with either 5 or 10 mM cysteine, the absorbance increased to 0.30 after a 21-day lag with a doubling time of 19 h, or to 0.38 after a 10-day lag with a doubling time of 26 h, respectively (Figure 2C, Table S2)."

Lines 542-545 were rewritten as: "For these experiments, uninoculated medium was tested for the abiotic formation of sulfide with the methylene blue test⁶¹. Because the test was inhibited by high concentrations of cysteine and dithiothreitol, the cysteine concentration was reduced to 1 mM and dithiothreitol was omitted. After 6 days at 37°C, no sulfide was produced with a limit of detection of 1 μM ."

Line 212: Should the infinity symbol subscript be a perpendicular symbol?

Corrected. It came from a file.docx to file.pdf conversion.

Line 243: Table 1 indicates the structures are resolved at rather poor resolution and the precise orientation of the [FeS] cluster and ligand (line 250, Figure 3) is uncertain, but the text is careful not to overinterpret the results, so this is OK.

We did not make any changes.

Line 247: cite the PDB ID of LpLSD.

done

Line 257-260: This sentence could be rewritten for better readability.

rephrased

Lines 267 and 309: A comment would be appropriate to explain why the occupancy of [4Fe-4S] cluster is 65% for holo-MmCyuA, but it is fully occupied for holo-C289A-MmCyuA.

We do not know why the occupancy of the [4Fe-4S] cluster is 65% for holo-MmCyuA and 100% for holo-C289A-MmCyuA. We guess that it comes from the cluster reconstitution step which is poorly reproducible. We do not think that this point deserves a comment in the text.

Line 289: Any explanation for why Holo-C289A-MmCyuA shows greater activity than holo-MmCyuA (Fig.2A)?

We do not think that the curves show a major difference between the activities of holo-C289A-MmCyuA. The slightly enhanced activity of the variant compared to the wild-type enzyme may be due to the faster binding of the cysteine substrate. Indeed, the cluster is more accessible in the variant due to the absence of the fourth cysteine Cys289 coordinating the cluster. This suggestion has been added to the text in the Discussion section.

Line 295: DNA is mutated, not proteins. Change “mutated to alanine” either to “mutated the codon for alanine” or “substituted with alanine”. The same issue occurs in lines 340, 405, 426.

corrected

Line 296: In Figure S14, change “mutants” to “variants”. Consider renumbering Table S4 to Table S3 to retain sequential numbering. Also, change “mutants” to “variants” in the title of the table.

Corrected

Line 370: “CyuA can be used as a carbon/energy source” might be true if the protein were degraded. The authors mean that “CyuA can metabolize L-cysteine as a carbon/energy source”.

corrected

Line 386: Define MjCyuA.

done

Line 414: Figure 5 suggests the protonated thiol group of substrate binds to the unique Fe of the [4Fe-4S] cluster and that the cluster-bound sulfide is protonated. No experimental evidence was reported to address the protonation states. The authors are encouraged to indicate the deprotonated species may alternatively occur in both cases. We agree that we do not have experimental evidence about the protonation states in Figure 5 (New Figure 6). Concerning the Cysteine substrate, we have now indicated in the Figure Legend that it could be protonated or deprotonated. We think that the formulation [4Fe-4S]-SH is the most appropriate for sulfide bound to the cluster

Line 513: Fix the typo (“probalbly”). Also, this paragraph has nine uses of “has been” or “have been” that could be simplified to “was” or “were”. The same issue is found elsewhere throughout the text.

done

Line 585: replace “BSA” with “bovine serum albumin”.

done

Line 609: use uppercase for 40K.

done

Reviewer #2 (Remarks to the Author):

I co-reviewed this manuscript with one of the reviewers who provided the listed reports. This is part of the Communications Biology initiative to facilitate training in peer review and to provide appropriate recognition for Early Career Researchers who co-review manuscripts.

Reviewer #3 (Remarks to the Author):

The article "Evolution, structure and function of L-cysteine desulfidase, an enzyme involved in sulfur metabolism in methanogenic archaea" authored by Sylvain Gervason, Paolo Zecchin, Elliot B. Shelton, et al. reports the structure and function of a protein called CyuA (*E. coli* nomenclature). Cysteine desulfidases are very interesting proteins that are certainly worth to have a closer look at. From this point of view and with respect to the quality of this trans-disciplinary study combining evolutionary biology, extraordinary structural biology and biochemistry, this paper must certainly be published as soon as possible.

However, some minor and one major point should be addressed prior publication:

Minor points:

1) lines 141-147: The authors say that the archaeon *M. maripaludis* is usually cultivated in 3-5 mM Na₂S but still cultivated them in 2mM calling this "optimal" concentrations. Can the authors explain why they don't use 3-5 mM?

This statement was an error on our part. *Methanococci* are usually cultivated with 2 mM sulfide, and the text was rewritten to reflect this.

2) lines 154: *M. maripaludis* (MmCyuA): the authors may put the proteins name not in brackets. This may lead to misunderstanding.

This sentence was rewritten, see next comment.

3) lines 154-155: it is misleading that both, high and low levels, are described as "abundant".

This sentence has been rewritten as: "Altogether, the relatively poor growth of the mutant in the presence of both low and high levels of sulfide suggests that MmCyuA plays an important role in metabolism even when sulfide is not limiting growth."

4) line 159: can the authors give an example where we actually do find high concentrations in the environment?

We believe that *methanococci* would rarely, if ever, encounter high cysteine levels in their environment. In these experiments, high concentrations of cysteine were used to clarify the metabolism and regulation of CyuA. We do not believe that they are relevant to their ecology. No change was made.

5) lines 175-176: this strategy should have also been tested in the initial growth tests. If the authors have this piece of data, it would be nice to include it to the manuscript.

This strategy was employed with the low inoculum cultures, and some of the results for growth with cysteine plus low concentrations of sulfide are reported in Figure 1B. The results for cysteine alone was not described in detail because no growth was observed. To make this clearer, we elaborate by revising the text as: "Since both the wild-type and mutant strains failed to grow with 1-10 mM cysteine as the sole sulfur source using a low inoculum, we increased the load of the inoculum 100-fold to provide sufficient initial biomass to allow grow..."

6) line 178: "the abiotic formation of sulfide from cysteine was not detected". How did the authors measure that, is there data to show?

Please see the response to Reviewer 1 for Lines 178-181. This section was revised to reflect this concern.

7) line 195: “after treatment with dithionite and EDTA to remove the residual cluster (Figure S5C, 7 dotted line), SEC-MALS analysis of ‘apo-MmCyuA’ thus obtained indicated that it was mainly a dimer (Figure S5D).” Please rephrase.

rephrased

8) lines 226-228: the study would greatly profit from including affinity measurements with respect to the different amino acids using e.g. MST (microscale thermophoresis).

We did not perform affinity measurements of the substrate/product/inhibitor with the holo-proteins because it is not possible to do this experiment with MST under anaerobic conditions in the glovebox.

9) lines 257-258: are there inter-domain connections that allow "communication" between the two subunits such as found in cysteine desulfurases? The paper would benefit from a closer look into the structure.

In cysteine desulfurases, the dimer interface is necessary to create a suitable binding pocket for the PLP. In cysteine desulfidases, the dimer interface does not provide catalytic residues but rather forms a hydrophobic surface. We believe that we currently have no evidence to explain how the two subunits communicate with each other.

10) line 281: the authors state that the two molecules in the asymmetric unit were only "similar". Are there notable differences?

In fact, after superposition of the two molecules in the asymmetric unit, the rmsd was 1.1 Å for 332 aligned C α s, indicating that the two molecules are not so similar to each other (Figure S13A). When superposing only the catalytic domains of the two molecules, the rmsd was 0.094 Å for 155 aligned C α s, indicating that the N-terminal domains adopted different orientations in the two molecules. We have changed the Figure accordingly (Figure S13A) and changed the text as follows: “The crystals, obtained also in space group P2₁2₁2₁, diffracted at 2.4 Å resolution and possessed two molecules in the asymmetric unit (Table 1, crystal 2), which differed by the orientation of the N-terminal domains relative to the catalytic domains (Figure S13A).”

11) line 282: why is there a 2Fe/2S found when the authors co-crystallize the protein with an inhibitor? Does this have any functional meaning?

We currently have no explanation for the presence of a [2Fe-2S] cluster trapped in the crystals when MES from the buffer solution occupies the active site. [4Fe-4S] clusters

are highly sensitive metallic centers and have been shown to degrade into [2Fe-2S] clusters in numerous other instances. We do not believe that the observed [2Fe-2S] cluster is physiologically relevant, but rather that it likely results from cluster degradation during this experiment.

12) lines 277 and following: what is the benefit of the paragraph "The crystal structure of the [2Fe-2S]-MmCyuA in complex with MES confirms the location of an anion binding site"? It may be worth to combine this paragraph with the following one and shorten it. We wish to retain this paragraph, as this structure provides interesting information at a higher resolution (2.4 Å) than the others. We have therefore made no changes.

13) line 303: the authors state "several loops". Please, be more precise. There is no useful info in that statement, yet.

We analyzed in more details the differences between the dimers of holo-MmCyuA alone and holo-C289A-MmCyuA after superposition of the C-terminal catalytic domains of molecules A of both crystal structures. The Isqab program in CCP4 indicates that loops 217-236, 279-287, 323-333 and 353-365 have rmsd much higher than the mean. This information has been added to the text.

14) line 300: It would be interesting to soak the WT enzyme with the molecules found in the structure of the mutant enzyme as well as the substrate and the product. Is there any reason the authors did not do this?

We tried to soak wild-type holo-CyuA and the C289A variant with cysteine, serine or pyruvate but did not obtain diffracting crystals.

15) line 309: "Fe edge" do the authors mean "Fe K edge"?

corrected

16) line 314: the presence of hydrogen bonds is rather a suggestion at a resolution of 3.2 Å.

The resolution of the holo-C289A-MmCyuA/ethylene glycol crystal structure is in fact 3.74 Å. We agree that the hydrogen bonds are not well defined at this resolution. Before describing potential hydrogen bonds, we have now added the following text : ' Despite the low resolution of 3.74 Å of this structure, it is likely that '.

17) line 345: "The C-terminal domain of MmCyuA displays strong structural similarity with the catalytic domain of serine dehydratases." What is the benefit of this data for the manuscript? Does it need a separate paragraph? It may be incorporated as only one sentence in the paragraph "Cys23 and Glu25 of MmCyuA are critical for the cysteine desulfidase activity"

We think that it is interesting to compare the structures of cysteine desulfidase and serine dehydratase, which share a similar catalytic function. The superposition of the active sites of holo-C289A-MmCyuA and LpLSD showed that the cysteine substrate can be modeled in the first structure with a position similar to that of the terminal cysteine in the LpLSD structure. We also discuss the nature of the base abstracting the C α proton of the substrate in both cases. We thus would like to keep this information.

18) line 385: "first homogeneous anaerobic preparation": this is not fully correct! It was prepared aerobically and then reconstituted anaerobically. Please correct.

corrected

19) lines 436/437: rephrasing needed.

rephrased

Major point:

In lines 188-189 the authors state that they purified the protein aerobically and in line 391 they explain that under such conditions Fe/S clusters may be degraded. Why do the authors not purify the protein anaerobically? They have shown that they have the expertise to do so.

Notably, the degradation on one hand is very common under aerobic conditions and on the other hand the reconstitution of proteins with Fe/S clusters does not necessarily yield the physiological type of Fe/S cluster. In fact, very often during chemical reconstitutions non-physiological 4Fe/4S clusters are incorporated into proteins that in vivo do not carry such cluster types. It is therefore strongly recommended to purify the protein, both WT and mutants, anaerobically and test for physiological cluster content.

We performed the experiment asked by this Reviewer. The extracts of *E coli* cells expressing wild-type CyuA and the C289A variant were reddish, consistent with the overexpressed proteins containing an Fe-S cluster. However, purification of these proteins within the glove box led to rapid precipitation. Therefore, the obtained amounts of purified proteins were not sufficient for analysis by UV-Visible spectroscopy and for their Fe content. We believe that different conditions should be tested to improve the experimental conditions, but that it is beyond the scope of this manuscript. In particular, overexpression tests in the presence of a system overexpressing the FeS cluster biogenesis machinery might help to maintain the proteins in the holo-forms, which could reduce protein precipitation and enhance their Fe content.

For future projects and with respect to physiology, of course an overexpression and purification from an archaeal host would be the gold standard, since *E. coli* may even in vivo equip enzymes with the wrong type of cluster. However, setting up such a procedure may take some time and hence this is a suggestion for the next study.

We will take this advice into account for the next study.

Reviewer #4 (Remarks to the Author):

Gervason et al characterize the taxonomic distribution, evolution, biochemical function, physiological function, and structure/function of L cysteine desulfidase (CyuA). They show that CyuA was unlikely to be a property of LUCA and rather likely emerged in Bacteria, with LGT to Archaea. Heterologous expression of CyuA from MmS2 shows the enzyme catalyzes the decomposition of cysteine to sulfide/ammonia/pyruvate, suggesting a potential function in liberating S for cellular biosynthesis. This is important, as this helps to explain several phenomena in sulfur metabolism of past studies of MmS2. Through deletion studies, the authors then show that this enzyme plays a key role in sulfur metabolism in MmS2, in particular under low sulfide growth conditions. It is suggested that CyuA assists in providing protection against cysteine toxicity. They obtained a structure and showed that the 4Fe4S cluster is required for activity (I will not comment on the structural or protein mutation aspects of this work, as I am not an expert in these areas). Overall, the dataset is comprehensive and fills a knowledge gap in our understanding of S metabolism in anaerobes, in particular MmS2. Overall, I enjoyed reading this manuscript and have a few suggestions to further improve it and to clarify several important points in preparation for readers of *Comm. Biol.*

General Comments:

The abstract would benefit by including some statement regarding the broader relevance/significance of CyuA to the microbiology field. Perhaps including information on the taxonomic distribution among anaerobes and/or the evolution of the protein – is it recent if it was not a property of LUCA? Does it help explain the lack of cysteine desulfurase homologs in some *Methanococcales*. As it stands, a reader of the abstract may not realize the broader implications of the work and rather may think this is a study on a peripheral protein (it is not but the worry is there).

We have remodeled the abstract to include a sentence about CyuA evolution in order to indicate to the reader the broad aspect of our work. We now also make a specific mention of *Methanococcales* in the results and discussion sections.

Concerning the lack of cysteine desulfurase in *Methanococcales*, we investigated their taxonomic distribution and compared it to the CyuA distribution (New Figure S3). The *Methanococcales* are an exception among CyuA-encoding prokaryotes as most of them possess a cysteine desulfurase. This suggests that the role of CyuA in *Methanococcales* could be potentially wider than in other organisms regarding the sulfur metabolism. This information was added in the 'Results' section of our revised manuscript.

I appreciated the authors of this largely biochemical study using phylogenetic/evolutionary approaches to make their work of broader importance, yet I was a bit underwhelmed with the treatment. More insight could be gleaned from the evolutionary analysis, including specific mention of where *Methanococcales* obtained CyuA homologs from.

We now make the message clearer that *Methanococcales* obtained CyuA through horizontal gene transfer from bacteria.

There is so much more hiding in that phylogeny in the SI that could be used to further the relevance of this work. I would even go so far as to suggest that the phylogeny (perhaps a collapsed version) should be included as primary figure in the paper, with combining other figures in the paper to keep within Comm. Biol. requirements.

Following this Reviewer's comment, we have now added the phylogeny of CyuA (Ancient Figure S3) as a main Figure (New Figure 1).

The authors often use extremely high concentrations of substrates in their assays (e.g., 10 mM cysteine) and then use the results of such assays to suggest ecological/physiological roles for CyuA in host organisms. It is hard for me to imagine an environment with concentrations of cysteine that approach the levels used herein and thus suggest such arguments be tempered.

We believe that *methanococci* would rarely if ever encounter the high cysteine levels used in these experiments in their environment. The high concentrations of cysteine were used to clarify the metabolism and regulation of CyuA and not to infer ecological roles. No change was made.

Further, the authors should consider whether the high sulfide numbers are accurate or if those studies they cite (>10 mM sulfide in salt marsh sediments) might be attributable to sulfide complexed with metals that would render the sulfide either non-toxic or not available. In other words, if there is really 10 mM of free sulfide, then the cells have major problems as nearly all metals will be precipitated as sulfide minerals. Alternatively, a fraction of what was called "free sulfide" may be acid dissociable soluble metal sulfide clusters that present as free sulfide in typical methylene blue assays (add acid to dissociate metal sulfide clusters before the assay).

While it may be true that much of the sulfide in salt marshes, the habitat for this methanogen, is complexed to metals, it really is not relevant to our study. Under our growth conditions, the concentration of sulfide far exceeds that of iron and trace minerals, such as molybdate. Certainly, *methanococci* grow in much higher sulfide concentrations than other methanogens and many other anaerobes, which was the point of this section. No change was made.

Specific comments:

Line 71: MmS2 has been shown previously to grow on cysteine alone or in combination with thiosulfate (see doi: 10.1038/s42003-024-07049-w). This does not take away from the current paper but it should be mentioned in the paper. Also, somewhere there is a study that showed that Methanococcales can generate sulfide from cysteine despite lacking cysteine desulfurase homologs. This is the paper that comes to mind: doi: 10.1074/jbc.M110.152447.

We were unable to reproduce the results of Payne et al. 2024 (10.1038/s42003-024-07049-w), and this was mentioned on lines 465-470 in the original manuscript. We stated that it was possible for there to be strain differences. Even though we both worked with strain S2, there are many sequence differences in laboratory strains of S2 (see Long et al. 2021; doi: 10.1099/acmi.0.000244). The reviewer also misread the paper Liu et al. 2010 (doi: 10.1074/jbc.M110.152447). Cysteine was not shown to replace sulfide as a sulfur source in that work. No change was made.

Line 97: Several members of the Methanococcales do encode SufS (please see Table 1 in doi: 10.1128/JB.00117-2). MmS2, as the authors correctly note, does not.

Thank you for calling this paper to our attention. We now cite Johnson et al. 2021.

Line 108: A quick read did not convince me that reference 26 actually made this claim and I would be surprised if this were true (which the authors then show to be the case later in their report). I would refer to Weiss 2016 (doi: 10.1038/nmicrobiol.2016.116) to see if they also pinned CyuA to LUCA (I do not believe they did) and use this work. This would also help clear up the confusion that resulted when I read this sentence and then read later that CyuA was not in LUCA.

In Reference 26, the authors said “This distribution may suggest an origin for CyuA that predates the split between archaea and bacteria.” which suggests that they refer to the presence of CyuA in the LUCA. In Weiss, 2016, the authors indeed did not infer CyuA as present in the LUCA. We thus included this information in the text :“This is consistent with a global analysis of the hypothetical genome of the LUCA, in which CyuA was not inferred as ancestral (Weiss, 2016).”

Line 124: What was the evolutionary path of CyuA among Archaea? How did Methanococcales end up with this protein? Additional detail would be appreciated here. We have added information about the emergence of CyuA in *Methanococcales* from horizontal gene transfer from bacteria, in the results and discussion sections.

Lines 144-145: Sulfide is toxic to MmS2 and cells actually prefer to grow with sulfide that is complexed with other metals such as Fe. It is also true that sulfide results in metal limitation for cells. Please see Fig. 1 in doi: 10.1038/s42003-023-05163-9. Further, cells grown with cysteine + thiosulfate grew nearly as well as those provided with sulfide (2 mM) indicating that exogenous sulfide is not necessary for optimal growth (see again Payne 2024). I suggest that the sentence be rephrased for accuracy.

While we agree that sulfide can inhibit growth under certain conditions, as shown elegantly by Payne et al. 2023, we do not feel that sulfide is toxic in the sense that it poisons metabolism. Certainly, under our conditions, 2 mM sulfide is not toxic and is required for good growth. Nevertheless, the text was modified to: "Accordingly, we observed that 2 mM concentrations of Na₂S were required for good growth of *M. maripaludis* under our conditions (Figure 1A)."

Lines 161-168: These concentrations of cysteine are not terribly relevant from a cytotoxicity standpoint (where would MmS2 encounter such high concentrations?). I wonder if the effect is attributable to such high concentrations of cysteine making it difficult for the cells to obtain the metals that they need (complexes the metals). Alternatively, is it possible that the role of cysteine in these results is to decrease the ORP of the medium or to scrub out O₂ when insufficient sulfide is available to do so?

We believe that *methanococci* would rarely if ever encounter high cysteine levels in their environment. In these experiments, high concentrations of cysteine were used to clarify the metabolism and regulation of CyuA and the mode of inhibition was not investigated. No change was made.

Line 495: This is a quite high e value and it is not clear how this value was settled on. I wonder if the authors, now that they have identified active site residues in CyuA, can return to their alignments and show retroactively that the previously identified CyuA homologs are likely to be these proteins? This would also provide a mechanism to validate (and ensure readers) that this e value is appropriate. And it would be a neat way to show how useful the combined biochemical/phylogenetic approach used in this study is and would provide a model for future studies of this type.

We used such high e-value to retrieve exhaustively the homologues, even for paralogues. Next, we inferred a preliminary phylogeny with the three paralogues and, we delineated the CyuA family according to phylogenetic distance with the other families, the number of copies per genome, the difference of length of sequences, annotations, and specific genomic contexts. We clarified the Material & Methods section.

We already mapped the five most important catalytic residues (the three conserved cysteines that coordinate the [4Fe-4S] cluster and the cysteine and glutamate residues involved in C α proton abstraction of the cysteine substrate) onto the phylogeny of CyuA (Ancient Figure S3 changed to New main Figure 1). Nevertheless, following the reviewer's advice, we now mapped these residues onto a tree of the complete protein family (containing the methyl-accepting chemotaxis proteins, CyuA and SdaAB) (New Figure S1). We can see that none of these residues are conserved within methyl-accepting chemotaxis protein, that the three cluster-coordinating cysteines are common to both CyuA and SdaAB enzymes, but not the two residues involved in proton abstraction. This analysis supports our phylogenetic delineation of CyuA enzymes. We now mention this information both in the Results section (when comparing CyuA and SdaAB enzymes) and in the discussion section. We thank the reviewer for this interesting question.

Congratulations on a nice study: Eric Boyd.

Thank you for the positive comment!

Reviewers' comments:

Reviewer #1 (Remarks to the Author):

The authors have adequately addressed the minor concerns I indicated in the original manuscript. Nice work!

We thank Reviewer 1 for his positive comment.

Reviewer #2 (Remarks to the Author):

I co-reviewed this manuscript with one of the reviewers who provided the listed reports. This is part of the Communications Biology initiative to facilitate training in peer review and to provide appropriate recognition for Early Career Researchers who co-review manuscripts.

Reviewer #3 (Remarks to the Author):

I am very happy with the answers given and changes made by the authors.

However, I strongly disagree that anaerobic MST was not possible. It was shown several times that it is well possible. E.g. refer to the works of Roland Lill. In fact it very simple since oxygen does not diffuse rapidly inside the capillary.

<https://resources.nanotempertech.com/application-notes/anaerobic-microscale-thermophoresis-reveals-the-redox-dependency-of-ferredoxin-in-mitochondrial-fe-s-biogenesis>

Since the exact affinity, however, does not improve the manuscript significantly, I think the manuscript should now be published in its present form as soon as possible.

We apologize. We did not know that anaerobic microscale thermophoresis was possible. We thank the Reviewer for acknowledging that this additional experiment is not essential for the acceptance of the manuscript.

Reviewer #4 (Remarks to the Author):

In general, the authors have adequately addressed my concern with the following exception(s): In my previous comment, I indicated the existence of "...a study that showed that Methanococcales can generate sulfide from cysteine despite lacking cysteine desulfurase homologs. This is the paper that comes to mind: doi: 10.1074/jbc.M110.152447". The reviewers state the following in response

The authors response is "The reviewer also misread the paper Liu et al. 2010 (doi: 10.1074/jbc.M110.152447). Cysteine was not shown to replace sulfide as a sulfur source in that work. No change was made."

This reviewer did not misread the paper. Rather, the authors misread the paper and my comment. As stated in Liu et al., 2020: "Although cysteine desulfurase homologs have not been identified in the *M. maripaludis* genome, cysteine desulfurase activity was detected in cell-free extracts." This suggests an alternative mechanism (not canonical cysteine desulfurase) to metabolize cysteine. The authors should adjust their description of this finding where necessary throughout the paper.

In fact, this reference (Reference 17) was already mentioned in the paper, and it was indicated that CyuA could account for the cysteine desulfurase activity reported in that paper.

In the initial comment, we thought the reviewer was discussing the activity of whole cells, which was not shown in the cited paper [which we wrote, reference 17]. However, it is correct that the paper reported that cysteine desulfurase activity was detected in cell extracts. We also pointed out that the assay was not specific and likely detected another enzyme capable of cysteine degradation.

We have rephrased and simplified lines 73-75 p. 3 to make this information clearer:
"Such cysteine desulfidase activity could account for the observed release of free sulfide from cysteine in cell extracts, reported previously in *M. maripaludis*¹⁷."

We did not find other places in the manuscript where it was necessary to add this information.

Secondly, if the authors seek to perpetuate false claims of the presence of CyuA in the ancestor of Bacteria/Archaea (e.g., LUCA) based on a weak study, it is their prerogative to do so. However, it would be cleaner to state that it was not a property of LUCA based on the more robust, recent study of Weiss and leave it at that. Especially since the author's own analyses (CyuA in methanococcales via HGT from bacteria) concur with Weiss.

We now refer to Weiss reference (New Reference 33) from the beginning of the manuscript.

- The first sentence of the Results section has now been changed to

'Previous phylogenetic analysis suggested that CyuA was not present in the Last Universal Common Ancestor (LUCA)³³

- 'In the second § of the Discussion section, we have now changed the sentence to

'Our phylogenetic results clearly confirm that most organisms possessing CyuA are anaerobes, supporting a role in anaerobic metabolism, and that it is not an ancestral protein, in agreement with a previous global analysis of the hypothetical genome of the LUCA³³.'